

# Estimation of diurnal emissions of CO₂ from thermal power plants using spaceborne IPDA lidar

Xuanye Zhang[1], Hailong Yang[2], Lingbing Bu[1], Zengchang Fan[1], Wei Xiao[3], Binglong Chen[4], Lu Zhang[4], Sihan Liu[5], Zhongting Wang[5], Jiqiao Liu[6], Weibiao Chen[6] and Xuhui Lee[7]

[1]School of Atmosphere Physics, Nanjing University of Information Science & Technology, Nanjing, 210044, China
[2]Shanghai Satellite Engineering Research Institute, Shanghai, 201109, China
[3]Nanjing University of Information Science and Technology, Yale-NUIST Center on Atmospheric Environment, Nanjing, 210044, China
[4]Key Laboratory of Radiometric Calibration and Validation for Environmental Satellites, National Satellite Meteorological Center (National Center for Space Weather), China Meteorological Administration. Innovation Center for FengYun Meteorological Satellite (FYSIC), Beijing, 100081, China
[5]Satellite Application Center for Ecology and Environment, Ministry of Ecology and Environment, Beijing, China
[6]Key Laboratory of Space Laser Communication and Detection Technology, Shanghai Institute of Optics and Fine Mechanics, Chinese Academy Of Sciences, Shanghai, China
[7]Yale University, School of the Environment, NewHaven, CT, America

*Correspondence to*: Lingbing Bu (lingbingbu@nuist.edu.cn)

**Abstract.** Coal-fired power plants are a major source of global carbon emissions, and accurately accounting for these significant emission sources is crucial in addressing global warming. Many previous studies have used Gaussian plume models to estimate power plant emissions, but there is a gap in observation capabilities for high-latitude regions and nighttime emissions. However, large emitting power plants exist in high-latitude areas. The DQ-1 satellite is equipped with the world's first active remote sensing lidar for detecting $CO_2$ column concentrations, which, compared to passive remote sensing satellites, enables observations in these regions. This paper applies a two-dimensional Gaussian plume model to the $XCO_2$ results from the DQ-1 satellite and analyses the instantaneous $CO_2$ emissions of 10 power plants globally. Among these, 15 cases of data are from nighttime observations, and 3 cases are from power plants located above 60° N latitude. The estimation results show good consistency when compared with emission inventories such as Climate TRACE and Carbon Brief, with a correlation coefficient R = 0.97. The correlation coefficient between the model fits and satellite observations ranges from 0.49 to 0.88, and the overall relative random error in the estimates is 15.11 %. This paper also analyses the diurnal and seasonal variations in $CO_2$ emissions from the power plants, finding that emission variations align with changes in electricity consumption in the surrounding regions. This method is effective for monitoring the diurnal variations of strong emission sources like power plants.

## 1 Introduction

Global warming is caused by the continuous increase of greenhouse gases in the atmosphere. The Kyoto Protocol under the United Nations Framework Convention on Climate Change classifies six gases, including carbon dioxide ($CO_2$), methane ($CH_4$), nitrous oxide ($N_2O$), hydrofluorocarbons (HFCS), perfluorocarbons (PFCS), and sulfur hexafluoride ($SF_6$), as major





greenhouse gases, with $CO_2$ being the largest contributor and a key anthropogenic greenhouse gas (Protocol, 1997). Changes in atmospheric composition due to industrial development, land-use changes, deforestation, and livestock farming have led to global warming and a series of severe events impacting the Earth's ecological environment, such as frequent natural disasters. These effects are further exacerbated by increasing greenhouse gas emissions (Arias et al., 2021; Searchinger et al., 2018). Currently, greenhouse gas emissions are accelerating, with global annual $CO_2$ emissions rising from 27 Pg to 49 Pg over the

past 40 years (Friedlingstein et al., 2022). In response to the severe challenges posed by climate change, countries worldwide are actively participating in $CO_2$ growth control initiatives and formulating strategies. China aims to peak $CO_2$ emissions by 2030 and achieve carbon neutrality by 2060 to curb the sharp rise in atmospheric $CO_2$ concentrations (Li et al., 2022).

Effective control of $CO_2$ emissions relies on accurate, timely, and transparent monitoring. Currently, countries assess emission reduction measures through greenhouse gas inventories, but challenges such as data lag, inconsistent standards, and insufficient

information transparency undermine comparability and credibility (Tubiello et al., 2015; Peters et al., 2012). For monitoring urban $CO_2$ emissions, most methods employ emission models based on inventory data, following a "bottom-up" approach (Gurney et al., 2017; Turnbull et al., 2018; Lauvaux et al., 2016). The WRF-STILT model is one of the most widely used atmospheric transport models for simulating urban greenhouse gas concentrations, but its performance in nighttime $CO_2$ simulation is hindered by the lack of emission height information in inventories (Turner et al., 2020; Pillai et al., 2012; Hu et

al., 2022). For small point sources like power plants, airborne or ground-based monitoring is typically used to measure $CO_2$ concentrations. Relevant studies have employed the mass balance method to assess $CO_2$ emissions from power plants and cities through airborne observations (Ahn et al., 2020). Some research teams have also utilized the inverse Gaussian plume model with MAMAP instruments to remotely sense the column-averaged dry-air mole fractions of $CO_2$ ($XCO_2$) from power plants (Krings et al., 2018; Krings et al., 2011). Ground-based equipment, such as portable Fourier transform spectrometers

(EM27/SUN), combined with the Gaussian plume model, has also been used to measure ground-level $CO_2$ concentrations for specific power plants and urban areas (Ohyama et al., 2021).

Satellite remote sensing technology holds significant potential for monitoring atmospheric $CO_2$ due to its capability for long-term, periodic observations on a global scale (Zhang et al., 2021). Monitoring point source emissions using satellites is challenging. Although the accuracy of the Gaussian plume model (GPM) is highly influenced by the precision of atmospheric

background fields (Nassar et al., 2017), it is not constrained by the spatial resolution of the model and remains stable and effective in simulating point source dispersion (Toja-Silva et al., 2017; Schwandner et al., 2017). The Orbiting Carbon Observatory-2 (OCO-2) is widely used due to its high measurement accuracy and stable results (Sheng et al., 2023; Crisp et al., 2017; Miller et al., 2007). When it passes downwind of a single point source, a significant increase in $XCO_2$ can be observed due to strong $CO_2$ emissions, and by fitting the observed $XCO_2$ with plume simulations, instantaneous $CO_2$ emissions can be

quantified (Nassar et al., 2017). In recent years, a series of studies have been conducted to estimate $CO_2$ emissions from power plants, volcanoes, and cities based on OCO-2's $XCO_2$ data (Nassar et al., 2017; Guo et al., 2023; Zheng et al., 2020; Crisp et al., 2017). Nassar et al. used the Gaussian plume model to estimate $CO_2$ emissions from 20 power plants and related facilities in the U.S., India, South Africa, Poland, Russia, and South Korea, noting an average difference of 15.1 % between the estimated



emissions and reported values for U.S. power plants (Nassar et al., 2021). However, OCO-2/OCO-3/GOSAT are passive
remote sensing satellites, which present data gaps in high-latitude and nighttime observations, and their spatial resolution of
approximately 3 km poses limitations for monitoring small-scale strong point sources (Shi et al., 2023; Taylor et al., 2022;
Eldering et al., 2019). For power plants, due to variations in power demand, nighttime emissions differ significantly from
daytime levels, and there may be instances of illegal nighttime over-emissions at certain plants, making nighttime $CO_2$ emission
observations necessary (Letu et al., 2014).

In 2022, China launched the Atmospheric Environment Monitoring Satellite (AEMS, also known as DQ-1), the first equipped
with spaceborne IPDA (Integrated Path Differential Absorption) lidar, capable of global $XCO_2$ measurements. This technology
addresses the gap in $XCO_2$ observations at high latitudes and during nighttime (Cai et al., 2022; Fan et al., 2024). Additionally,
the $XCO_2$ measurement accuracy is less than 1 ppm, with a footprint interval of 330 meters, significantly enhancing spatial
and temporal coverage for power plant monitoring (Zhang et al., 2024; Zhang et al., 2023). Han et al. applied the EMI-GATE
model, based on the Gaussian plume, to evaluate power plant emissions using DQ-1 data (Han et al., 2024). Compared to Han
et al.'s research, this study employs a different approach to the Gaussian plume model, resulting in lower random errors in
emission estimates. This paper also analyses two years of satellite data, examining emission variations of a single power plant
over time. Section 2 introduces the data sources and methods of this study. In Section 3, the improved Gaussian plume model
is integrated with DQ-1 satellite observations, selecting 10 globally high-emission power plants, including 15 nighttime
observations and 23 observations of power plants in high-latitude regions, estimating their $CO_2$ emissions. Analyses of diurnal
and seasonal variations in $CO_2$ emissions are also conducted. Section 4 provides a summary and discusses the potential
applications of the Gaussian plume model with spaceborne IPDA lidar.

## 2 Data and Methodology

### 2.1 Data

#### 2.1.1 DQ-1 satellite data

On 16 April 2022, China launched the world's first satellite designed for active remote sensing of carbon dioxide. This satellite
is equipped with an Aerosol and Carbon Dioxide Laser Detection Lidar (ACDL). The primary scientific objectives are to
measure high-resolution vertical profiles and the optical properties of global atmospheric aerosols and clouds, as well as to
obtain global atmospheric $CO_2$ column concentration data. This provides precise quantitative information for studies on $CO_2$
sources and sinks (Fan et al., 2024). The satellite utilizes Integrated Path Differential Absorption (IPDA) technology to measure
$CO_2$ column concentration. It employs a 1572 nm pulsed laser and the IPDA lidar method, using two wavelengths (λon and
λoff, corresponding to regions of strong and weak absorption lines). The difference in absorption cross-sections (σ) between
these two wavelengths is used to determine the $CO_2$ column concentration. On-orbit tests of the lidar have yielded high-
precision remote sensing data, confirming that the $CO_2$ column concentration measurement accuracy is better than 1 ppm.



Notably, this satellite provides the first global $CO_2$ column concentration measurements at night and over the poles (Zhang et al., 2024). The satellite's footprint spacing of 330 meters ensures high spatial resolution. This study utilizes the satellite's L2D product, which includes global $XCO_2$ data derived from raw observation data combined with the IPDA lidar inversion method. The datasets include $XCO_2$ values, uncertainty for $XCO_2$, and the corresponding surface elevation and geographic coordinates for each footprint.

**2.1.2 Wind data**

This study utilizes horizontal (U) and vertical (V) wind components from the fifth-generation European Centre for Medium-Range Weather Forecasts (ECMWF) global climate atmospheric reanalysis dataset (ERA-5). The dataset features a temporal resolution of 1 hour, a spatial resolution of 0.25°, and includes 37 vertical pressure levels (Hersbach et al., 2020). A four-dimensional interpolation method is applied to the U and V vectors at the plume lift height, which is set to a default chimney 110 height of 240 meters and a plume vertical lift height of 250 meters (Hu and Shi, 2021). To evaluate the impact of wind speed uncertainty on power plant emission predictions, the study also compares results using MERRA-2 horizontal wind data (Gelaro et al., 2017). Ground-level wind speed data are selected from the ERA5-land hourly surface wind speed U and V vectors, which are spatially interpolated. Additionally, the water vapor column content is derived from the spatial and temporal interpolation of the water vapor column concentration in ERA5-land.

**2.1.3 $CO_2$ emissions data**

The power plant validation data used in this study are sourced from the Carbon Brief database (https://www.carbonbrief.org/mapped-worlds-coal-power-plants/). However, many power plants in high-latitude regions do not provide emission data. Therefore, this study also compares the results with those from Climate TRACE (https://climatetrace.org/explore/electricity-generation-co2e100-2022). Climate TRACE estimates the activity levels (capacity 120 factors) of power plants and other facilities using satellite observations and machine learning methods. This database provides annual $CO_2$ emissions and power generation capacities for over 500 power plants worldwide.

**2.2 Emission inversion and Emission Uncertainties**

Gaussian plume models are widely used for monitoring point source emissions due to their stability (Brusca et al., 2016). This study applies this method to spaceborne IPDA lidar to estimate $CO_2$ emissions from power plants. The basic equation of the 125 model is as follows (Bovensmann et al., 2010):

$$\Delta Q(x,y) = \frac{F}{\sqrt{2\pi}a \cdot \left(\frac{x}{x_0}\right)^{0.894} \mu} e^{-\frac{1}{2}\left(\frac{y}{a \cdot \left(\frac{x}{x_0}\right)^{0.894}}\right)^2} \tag{1}$$



Where $x$ and $y$ represent the distances from the chimney along the wind direction and vertical to the wind direction (m), $\Delta Q$ is the total $CO_2$ column increment (g m$^{-2}$), $F$ is the point source $CO_2$ emission rate (g s$^{-1}$), and a is the atmospheric stability parameter, which is related to the solar radiation index and surface wind speed. The solar radiation index can be assessed using high cloud cover, low cloud cover, and solar elevation angle (Pasquill, 1961; Beals, 1971). The total $CO_2$ column amount converted to the increment of column concentration $\Delta XCO_2$ (ppm) can be calculated using the following equation (Zheng et al., 2020):

$$\Delta XCO_2(x,y) = \Delta Q(x,y) \cdot \frac{M_{air}}{M_{CO_2}} \cdot \frac{g}{P_{\text{surf}} - w \cdot g} \cdot 1000 \tag{2}$$

Where $M_{air}$ is the molecular mass of dry air (g mol$^{-1}$), $M_{CO_2}$ is the molecular mass of carbon dioxide (g mol$^{-1}$), $g$ is the acceleration due to gravity, $P_{\text{surf}}$ is the surface pressure (Pa), and $w$ is the water vapor column content (kg m²).

The satellite-observed $XCO_2$ results need to be converted into the $CO_2$ increment $\Delta XCO_2$ caused by power plant emissions. The diffusion of $CO_2$ plumes can be simplified using a two-dimensional Gaussian model, where the footprint of the spaceborne lidar is tangent to the two-dimensional Gaussian plume, leading to a shape similar to a one-dimensional Gaussian distribution. It is assumed that the background $CO_2$ concentration might exhibit a small gradient linear change, and $XCO_2$ distribution is considered to follow the distribution:

$$XCO_2(x) = XCO2_b + b \cdot x + \frac{a}{\sigma\sqrt{2\pi}} e^{[-(x-\mu)^2/2\sigma^2]} \tag{3}$$

Where $XCO2_b + b \cdot x$ is background value of $XCO_2$, $\frac{a}{\sigma\sqrt{2\pi}} e^{[-(x-\mu)^2/2\sigma^2]}$ is $\Delta XCO_2$ caused by power plant emissions (Reuter et al., 2019).





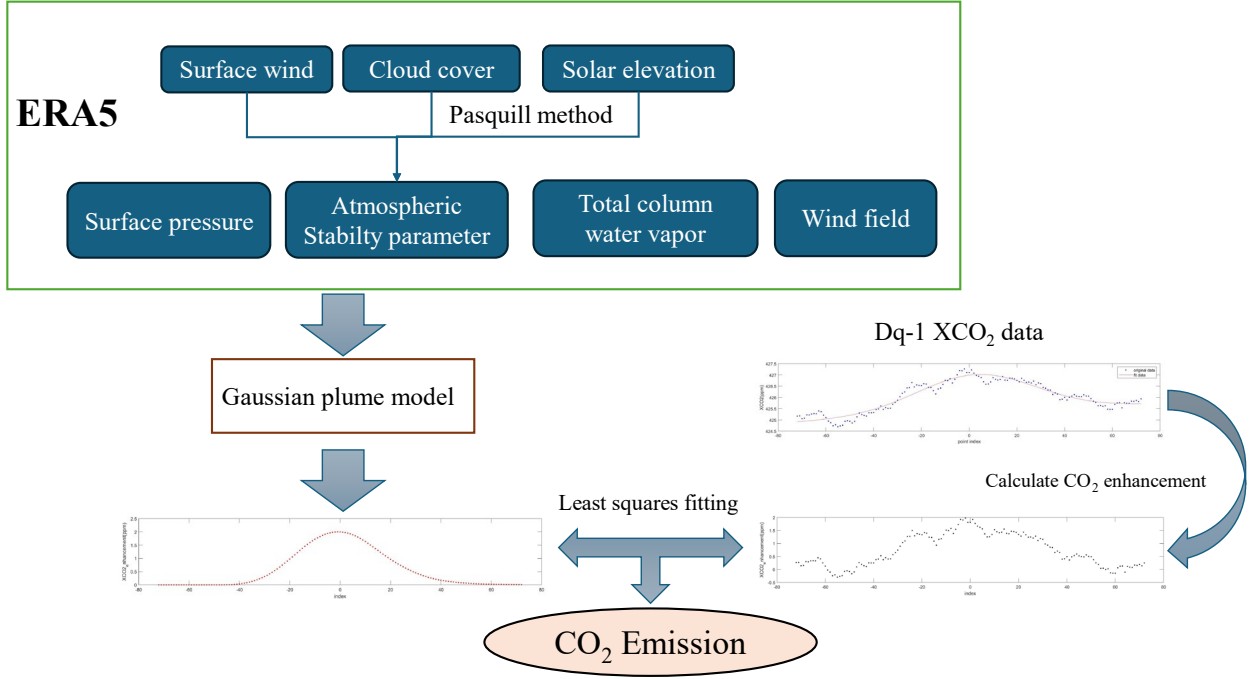

**Figure 1: inversion framework for Gaussian plume models.**

The specific calculation process is illustrated in Fig. 1. To improve the model's fit with satellite observation results, the selected DQ-1 orbital data require that the downwind direction of the point source intersects with the satellite footprint and that the distance between the $XCO_2$ enhancement location and the point source is less than 30 km. In the simulation, the x-axis of the Gaussian plume represents the direction of diffusion. Most studies use interpolated wind vectors from sources like ERA-5 at

the plume height as the plume's direction (Guo et al., 2023). However, in practice, wind direction may deviate and change continuously, making instantaneous wind direction insufficient for accurate plume propagation. In this model, the plume propagation direction is defined as the vector from the chimney location to the centre of the Gaussian peak. This direction is then compared with the interpolated wind direction. Only results where the wind direction difference is less than 25° are selected for further comparison and validation.

The selected satellite observation results are ultimately fitted to the model's plume results at the satellite footprint using the least squares method to obtain the $CO_2$ emission rate of the target power plant. The model's results can be calculated using Eq. (1) and (2), where the atmospheric stability parameter significantly affects the dispersion of the plume. Direct fitting with a specific value can easily lead to estimation bias. In this study, the atmospheric stability parameter is empirically interpolated by considering factors such as cloud cover and solar altitude. Then, the prior value of the atmospheric stability parameter is

varied by one standard deviation, and least squares fitting is performed accordingly. The optimal result is selected as the atmospheric instability of the location. For DQ-1 observational data, smoothing was applied to improve the signal-to-noise ratio and reduce errors (Zhang et al., 2023). Therefore, during the least squares fitting process, the model results are also



smoothed similarly. The primary differences between the Gaussian plume model used in this study and the EMI-GATE model employed by Han et al. lie in the calculation methods for the atmospheric instability parameter, background $CO_2$ column concentration, wind speed, and wind direction. Our approach involves varying these parameters within their respective error ranges based on the original observational values, with each parameter being calculated independently to maximize the interpretability of the results.

This study estimates the uncertainty in power plant emissions using a Gaussian plume model, considering five factors: uncertainty in wind speed, uncertainty in wind direction, uncertainty in plume height, uncertainty in atmospheric stability, and uncertainty in background field. The total uncertainty can be calculated using Eq. (4):

$$\varepsilon = \sqrt{\varepsilon_s^2 + \varepsilon_d^2 + \varepsilon_h^2 + \varepsilon_a^2 + \varepsilon_b^2} \tag{4}$$

Where $\varepsilon_s$ represents the error caused by wind speed. This is estimated by comparing the $CO_2$ emissions from the target power plant using wind speeds interpolated from MERRA-2 and ERA-5 data, with the wind speed uncertainty given by the difference between the two predictions. $\varepsilon_d$ represents the error caused by wind direction, calculated as the difference in $CO_2$ emissions using wind directions interpolated from ERA-5 versus the plume direction computed in this study. $\varepsilon_h$ represents the error caused by the emission height of the power plant. Assuming a default chimney height of 240 meters, and considering variations among different plants, including uncertainty in the plume rise height, scenarios with chimney heights of 160 m, 200 m, 240 m, 280 m, and 320 m are used to estimate the uncertainty. $\varepsilon_a$ represents the error due to atmospheric instability. The uncertainty due to atmospheric instability is calculated by assuming fluctuations of one standard deviation in atmospheric stability parameters. $\varepsilon_b$ represents the error in calculating the $CO_2$ background value. This is determined by comparing the average $CO_2$ concentration at points upwind of the source, outside the Gaussian plume, with the background value computed using the Gaussian fitting method employed in this study, thus providing the uncertainty in the $CO_2$ background field.

## 3 Results and Discussion

In this study, we utilized the DQ-1 satellite's Level 2D $XCO_2$ data and selected power plants with characteristics such as being located at mid-to-high latitudes and having large $CO_2$ emissions from Climate TRACE. We retrieved all satellite orbits passing within a circular area centred on the power plant chimney with a radius of 50 km, using ERA-5 wind direction data to filter the satellite orbits. For 10 typical coal-fired power plants, 47 sets of satellite footprints were found within the plant area. The Gaussian plume model was applied with stringent data filtering criteria, requiring no thick clouds and ensuring that the point source's downwind direction intersected with the satellite orbit. Under the condition that the error between wind direction and plume dispersion direction was less than 25°, a total of 28 cases were selected, including 15 nighttime observations and 3 cases where the power plants were located above 60° N latitude.



### 3.1 Emissions from high latitude power plants

Reftinskaya GRES (61.7° E, 57.1° N) is the largest solid fuel-fired power plant in Russia, generating electricity by burning coal. The plant emits not only $CO_2$ and other greenhouse gases but also large amounts of $SO_2$, $NO_x$, and other pollutants, making the monitoring of its emissions highly significant. Located in Sverdlovsk Oblast, the plant has a total installed capacity of 3,800 MW and produces 20 billion kWh of electricity annually. Climate TRACE data shows that its $CO_2$ emissions in 2022 were 22.7 Mt, ranking it 8th among global power plants. This plant is a major power source for the Sverdlovsk, Tyumen, Perm, and Chelyabinsk regions. The plant's Chimney No. 4, at 330 meters, is one of the tallest chimneys in the world, while the heights of the other chimneys are still uncertain. Due to the plant's high latitude, around 57° N, traditional passive remote sensing satellite data has a low efficiency (OCO-2 satellites have no valid Gaussian plume data for this plant), making it difficult to observe. However, active remote sensing methods provide high data coverage in high-latitude regions, allowing for more accurate estimates of the plant's emissions. In this study, we retrieved two years of observational data from July 2022 to July 2024, identifying a total of 27 valid satellite orbits passing over the plant, as shown in Fig. 2. Based on ERA-5 wind direction data and the $XCO_2$ distribution, 19 valid observations were identified, covering both daytime and nighttime during autumn/winter and spring/summer. These data enable analysis of the plant's emissions over time, with typical daytime and nighttime observation results presented in Fig. 3.

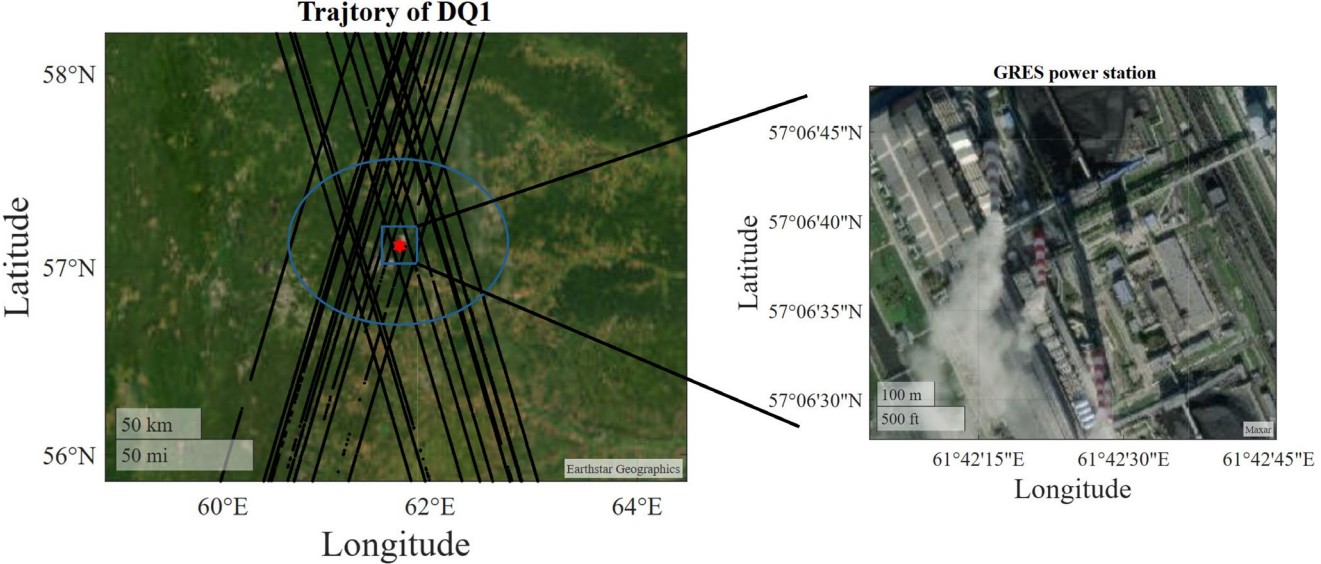

**Figure 2: The DQ-1 satellite passed through all orbits around the Reftinskaya GRES power plant, where the red hexagonal star indicates the position of the power plant.**





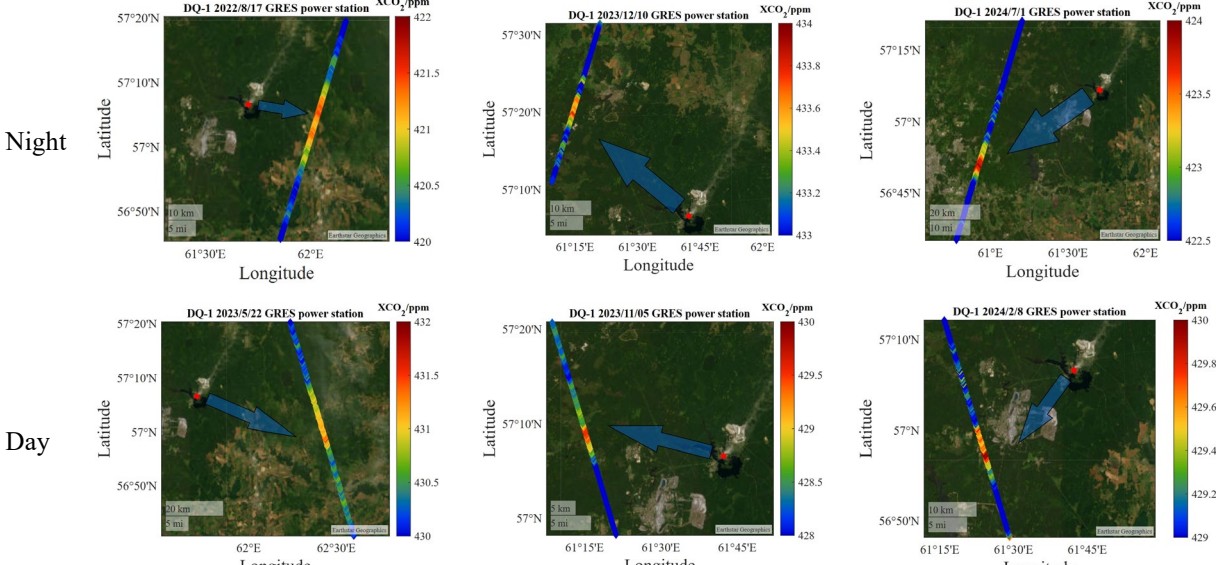

**Figure 3: July 2022-July 2024 DQ-1 satellite passes all orbits around the Reftinskaya GRES power plant, where the red hexagonal star indicates the position of the power plant.**

For the nighttime observation on 17 August 2022, the satellite orbit (Fig. 4a) was approximately 12 km from the power plant. Using Gaussian fitting, the background $CO_2$ column concentration was calculated to be 419.97 ppm, with an $XCO_2$ increment

of about 1.3 ppm (Fig. 4b). Within the plume, there were 76 data points, assuming a plume height of 530 meters above ground level (the sum of the chimney height and the assumed uplift height). The $CO_2$ emission rate of the inventory is 721.3 kg s$^{-1}$. Combined with the two-dimensional Gaussian model, the theoretical $XCO_2$ enhancement results were calculated (as shown in Fig. 4c), and using the least squares method, the fitted $CO_2$ emission result was 806.0 ± 108.2 kg s$^{-1}$, with a correlation coefficient of R = 0.88 (Fig. 4d). The average deviation between the model results and satellite-measured data was 0.32 ppm.

The total relative error of 13.4 %, which included an uncertainty of 70.5 kg s$^{-1}$ due to wind conditions, 39.8 kg s$^{-1}$ due to background levels, and uncertainties of 26.1 kg s$^{-1}$ and 64.4 kg s$^{-1}$ due to plume height and atmospheric stability, respectively. The slightly higher result from the model compared to the emissions inventory can be attributed to the fact that the emissions inventory represents annual averages. When converting these averages into instantaneous emission rates, the result tends to be lower than the actual instantaneous emission due to shutdowns for maintenance throughout the year. Although electricity

demand decreases at night in the mild summer climate, and the power plant's output is reduced, lower operational efficiency at low loads can lead to incomplete fuel combustion, resulting in overall $CO_2$ emissions slightly exceeding the emissions inventory (Hendriks, 2012).





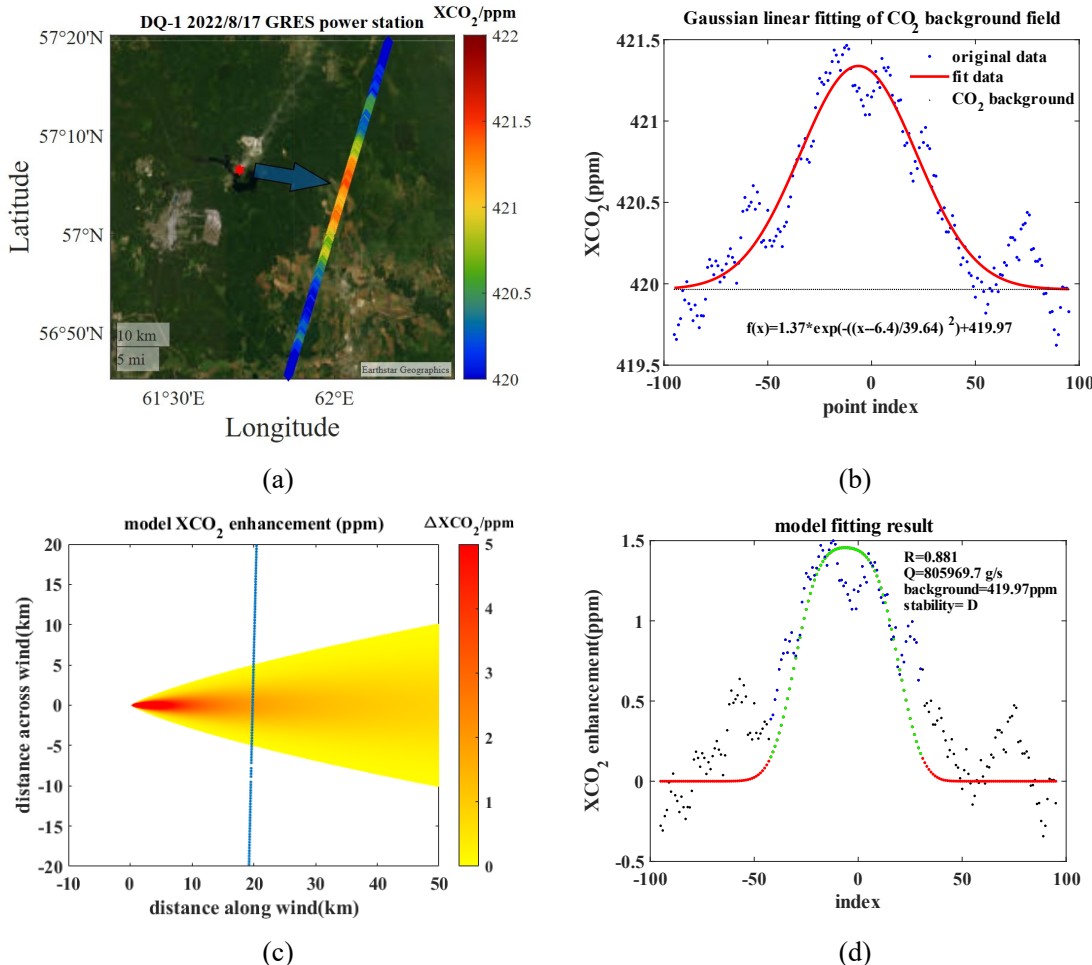

<p align="center">(a)</p>

<p align="center">(b)</p>

<p align="center">(c)</p>

<p align="center">(d)</p>

**Figure 4: (a) DQ-1 satellite observation on 17 August 2022, where the red six-pointed star indicates the location of the power plant,**
**and the red arrow indicates the result of wind interpolation of the height of the smoke plume at that location. (b)The result of one-**
**dimensional linear Gaussian fitting of the satellite observation of XCO₂ results, the red line is the fitted result. (c) Gaussian plume**
**distribution corresponding to the emission results calculated by the model, the blue point is the position of the satellite through the**
**plume. (d) Comparison between the XCO₂ enhancement results fitted by the model and the measured results, with the red and green**
**points indicating the model results, where the green points are the parts contained in the plume, and the black and blue points are**
**the measured results of the satellites, and the blue points are the points in the plume.**

On the night of 10 December 2023, the satellite also passed over this power plant, with the corresponding satellite trajectory

(Fig. 5a) located about 31 km from the plant. Using Gaussian fitting, the background value was determined to be 432.42 ppm,

and the XCO₂ enhancement was 1.2 ppm. There were 57 points within the Gaussian plume, and the Gaussian plume model

predicted the instantaneous emission rate of the plant to be $1027.5 \pm 177$ kg s$^{-1}$, with a correlation coefficient R = 0.87. During

the error calculation, the wind speeds from ERA-5 and MERRA-2 were 6.9 m s$^{-1}$ and 7.8 m s$^{-1}$, respectively, contributing an

uncertainty of 110.6 kg s$^{-1}$ due to wind conditions. Additionally, the atmospheric stability was calculated to be category D,

leading to an emissions uncertainty of 94.4 kg s$^{-1}$. Considering all factors, the total relative error was 17.3 %. Since December

<p align="center">10</p>





is already winter in Russia, the increased electricity demand for city lighting, transportation, and residential heating appliances (Savić et al., 2014) required the plant to maintain higher power output to meet the surrounding cities' electricity needs, leading

to an increase in instantaneous emissions.

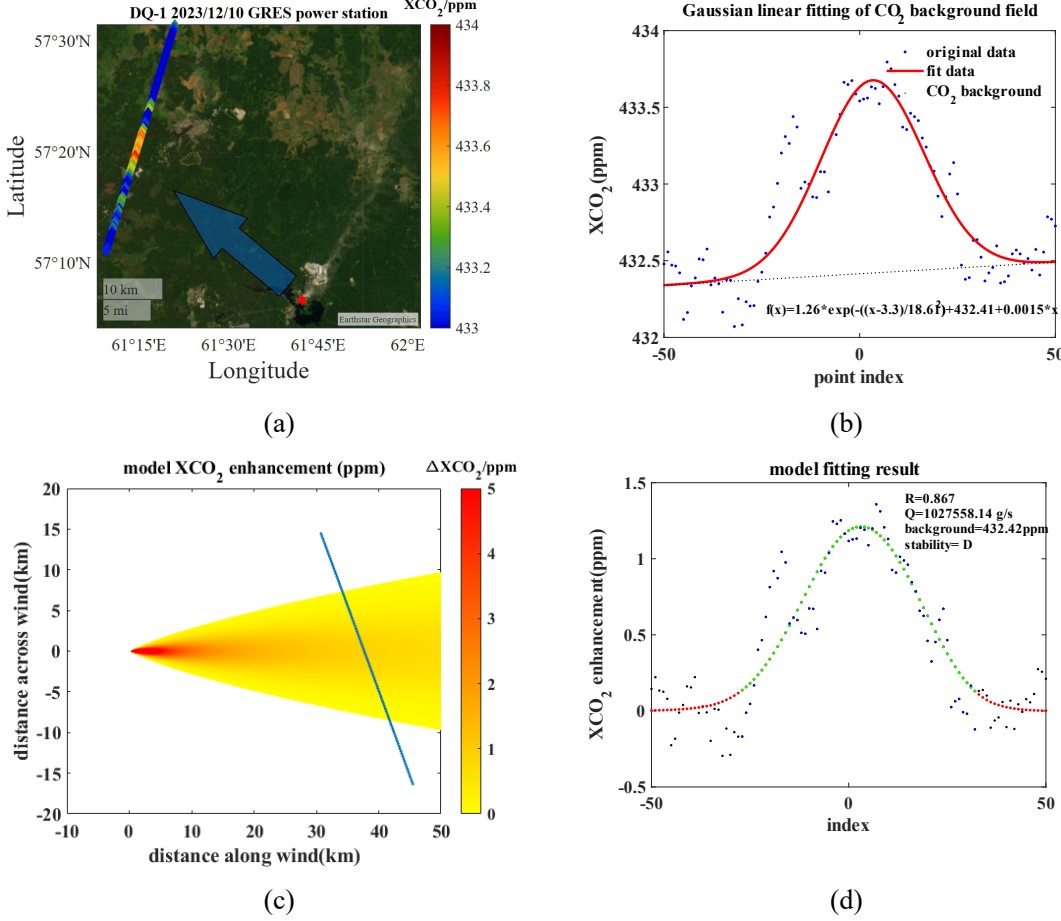

**Figure 5: (a) Observations from the DQ-1 satellite on 20 December 2023. (b)The result after fitting the satellite observation of XCO$_2$ results to a one-dimensional linear Gaussian, the red line is the fitted result. (c) Gaussian plume distribution corresponding to the emission results calculated by the model (d) Comparison of the model-fitted XCO$_2$ enhancement results with the observation results.**

On 8 February 2024, the DQ-1 satellite passed over the power plant again during the day at 08:29 UTC (Fig. 6a), with the Gaussian plume center located about 21 km downstream of the wind direction. In this observation, the atmospheric background field exhibited a strong linear variation trend, and the Gaussian linear fitting results are shown in Fig. 6b, with an average XCO$_2$ background concentration of 428.9 ppm. The surface wind speed was 4.2 m s$^{-1}$, and the atmospheric stability was calculated to be class B. There were 51 points within the plume. Using the Gaussian plume model, the plant's instantaneous

emission rates was predicted to be 1109 ± 169 kg s$^{-1}$, with a correlation coefficient R = 0.875 between the model results and satellite observations. The total uncertainty in the estimated emission rates was 168.9 kg s$^{-1}$, with a relative error of 15.3 %.





The largest contribution to this uncertainty was from the variation in the $CO_2$ background field, which caused an emission calculation uncertainty of 109.3 kg s$^{-1}$. The uncertainties due to wind field, plume height, and atmospheric stability were 88.9 kg s$^{-1}$, 31.5 kg s$^{-1}$, and 85.4 kg s$^{-1}$, respectively. Compared to the observation in December, the $CO_2$ emissions were higher in
this February observation, which can be attributed to the fact that the observation was conducted during the day when electricity demand is higher due to residents' work activities, leading the power plant to increase output, thereby raising $CO_2$ emissions (Waite et al., 2017).

**Figure 6: (a) Observations from the DQ-1 satellite on 8 February 2024. (b)Fitting the satellite observation of XCO₂ results to a one-**
**dimensional linear Gaussian, the red line is the fitted result. (c) Gaussian plume distribution corresponding to the CO₂ emissions calculated by the model (d) Comparison between the XCO₂ enhancement results fitted by the model and the measured results.**

By analyzing two years of observation data from the GRES power plant (as shown in Fig. 7), the overall estimated average emission rate is higher than the emissions reported in the inventory. The plant undergoes annual shutdowns for maintenance,





and the satellite observations represent instantaneous emissions, which may differ slightly from the annual average emissions.
Additionally, the Climate TRACE data reflects the 2022 annual average emissions, and the plant's yearly emissions fluctuate due to varying local electricity demand. Fig. 7 shows that summer emissions are lower than winter emissions. It was found that the plant is located in a high-latitude region where the climate is mild in summer and cold in winter. During winter, residents use electrical appliances and heating systems more frequently, and the power demand for urban infrastructure is higher than in summer. As a result, the power plant adjusts its output, leading to higher $CO_2$ emissions in winter (Savić et al.,
2014). The comparison between daytime and nighttime observations shows that the average $CO_2$ emission rate during the day is 1022 kg s$^{-1}$, while the nighttime average is 796 kg s$^{-1}$. The ratio of daytime to nighttime emission rates is 1.28. This ratio can be used to estimate the full-day $CO_2$ emissions when only daytime or nighttime observations are available. The power plant is the primary source of electricity for the region, and electricity demand from production activities during the day is much higher than at night. Consequently, the plant increases its load during the day, resulting in higher $CO_2$ emissions. This
also indicates that the plant does not engage in unauthorized nighttime emissions during the observation period.

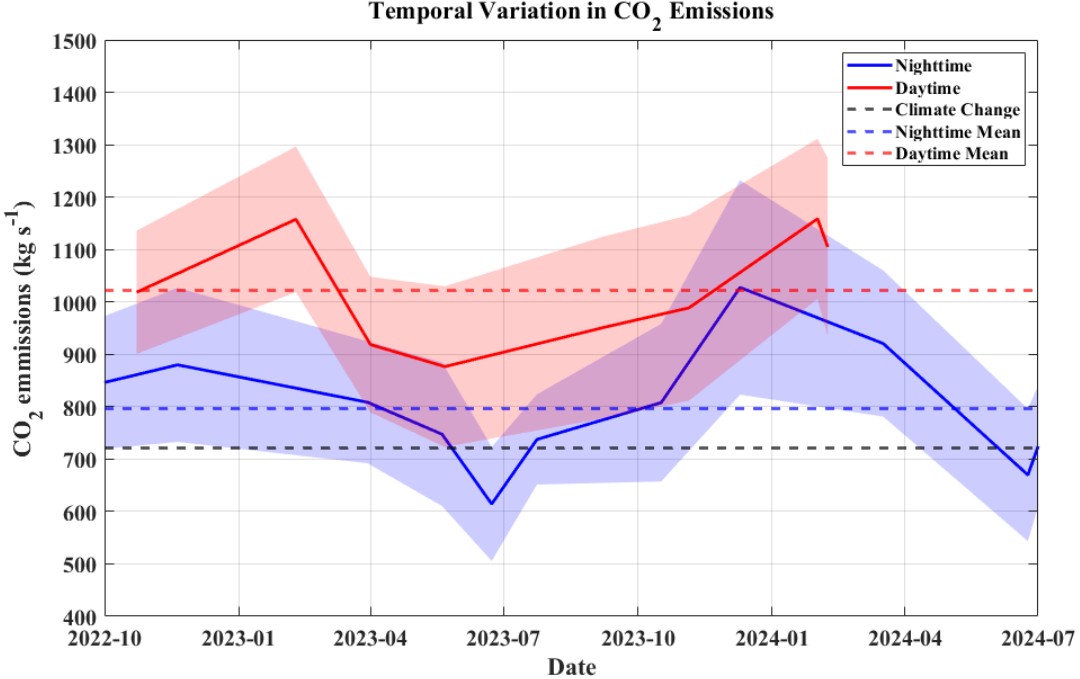

Figure 7: Diurnal emission rates of $CO_2$ from the GRES power plant over a 2-year period, with daytime results in red and nighttime observations in blue

## 3.2 Emissions uncertainty analysis

The uncertainty in the model calculations was assessed using Eq. (4) from Sect. 2.2, revealing that the uncertainty contributions vary across different power plants and influencing factors. The overall uncertainty results are presented in Table 1. The average relative random error is 15.11 %, which is lower than the 18.8 % random error of the EMI-GATE model (Han et al., 2024).



The primary contributors to this uncertainty are errors in the background field calculation, wind field errors, and atmospheric stability errors. Regarding wind fields, both the ERA-5 and MERRA-2 datasets are reanalysis results However, discrepancies can occur between these datasets, particularly in high-latitude regions where wind speed observations are sparse. When these wind speeds are used in the Gaussian model to estimate power plant emissions, differences can lead to errors in Gaussian diffusion velocity, thereby affecting prediction accuracy (Nassar et al., 2021). In this study, wind field-related uncertainty accounts for 26.7 % of the total error, with an average relative random error of 7.4 %. Atmospheric stability is another significant factor contributing to uncertainty. Atmospheric stability is not constant and varies in real-time with solar radiation, which is influenced by factors such as cloud cover and solar elevation angle (Ashrafi and Hoshyaripour, 2010). Since atmospheric stability directly impacts the shape of Gaussian diffusion, it introduces errors in predicted $CO_2$ emissions. For all results considered, uncertainty due to atmospheric stability contributes 25.1 % to the total error, with an average relative random error of 7.3 %. Plume height uncertainty also plays a role in the overall error. While some power plants provide chimney height data, allowing for the consideration of plume rise uncertainty only, others require an assumed chimney height (Guo et al., 2023). This assumption can lead to relatively high errors, primarily because wind speed and direction near the ground can vary with height, resulting in inaccuracies in wind field calculations. Plume height uncertainty contributes 6.5 % to the total error, with an average relative random error of 3.3 %. Uncertainty in the background field is mainly due to inaccuracies in its calculation. In areas with significant anthropogenic interference, a linear function may not adequately represent changes in background $CO_2$ concentration. Although the $XCO_2$ observation accuracy of the DQ-1 satellite is better than 1 ppm, uncertainty in its results contributes to errors in point source estimation. Background field errors account for 40.7 % of the total error, with an average relative random error of 9.5 %. For spaceborne lidar, the spatial distribution of satellite nadir points differs from that of passive remote sensing satellites, leading to fewer observed points within the plume. This increases the weight of background field uncertainty in the total error (Nassar et al., 2021; Shi et al., 2023).

**Table 1 Uncertainty caused by different error factors in the forecast results of different power plants.**

| Power Station | Wind field (Kg s$^{-1}$) | Plume height (Kg s$^{-1}$) | Stability (Kg s$^{-1}$) | Background field (Kg s$^{-1}$) | Total error (Kg s$^{-1}$) | Relative error |
|---|---|---|---|---|---|---|
| Scherer | 34.2 | 13.9 | 59.4 | 47.4 | 72.4 | 15.1 % |
| Belchatow | 48.3 | 14.5 | 72.9 | 98.8 | 134.2 | 15.4 % |
| Medupi | 64 | 45.8 | 12.3 | 52.8 | 98 | 16.4 % |
| Matimba | 81.3 | 21.7 | 34.8 | 75.5 | 118.3 | 16.7 % |
| CHP-1 | 4.3 | 1.9 | 9.5 | 15.2 | 18.4 | 16.7 % |
| CHP-3 | 5.2 | 1.8 | 5.8 | 9.1 | 12.1 | 21.2 % |
| GRES-2 | 43.2 | 31.2 | 72.3 | 107.2 | 143.5 | 12.6 % |
| GRES | 80.3 | 27.8 | 71.8 | 90.7 | 147.3 | 15.9 % |
| Taean | 51.8 | 27.4 | 10.8 | 27.8 | 73.3 | 7.4 % |
| Daesan | 2.8 | 0.9 | 1.6 | 1.2 | 3.5 | 12.7 % |



### 3.3 Validation of Emission results

The estimations from this study were compared with the emission inventories provided by Climate TRACE and Carbon Brief, as shown in Table 2. Both inventories present the annual total emission values, whereas the model results calculated from satellite observations represent instantaneous emission rates. Due to the power plant's ability to adjust its output based on local electricity demand, some differences between the two sets of results are expected. However, most results fall within the error range of the model predictions. The study includes three observation cases for latitude above 60° N and 15 cases of night-time emission detections. The use of spaceborne lidar to detect $XCO_2$ effectively compensates for the limitations of passive remote sensing satellites in high-latitude and night-time observations. A comparison of all observation results with the Climate TRACE inventory is shown in Fig. 8, with a correlation coefficient of 0.97.

**Table 2 Information on different power plants and the comparison of model predictions with emission inventories**

| Country | Station | Latitude | UTC Time | Model Result (kg s⁻¹) | Climate TRACE (kg s⁻¹) | Carbon Brief (kg s⁻¹) | Day or Night |
|---|---|---|---|---|---|---|---|
| Russia | GRES | 57.11° N | 8/17 22:12 | 806.0±108 | 721.3 | 638 | Night |
| Russia | GRES | 57.11° N | 5/22 08:35 | 876.2 ±153 | 721.3 | 638 | Day |
| Russia | GRES | 57.11° N | 11/05 22:10 | 988.6±161 | 721.3 | 638 | Night |
| Russia | GRES | 57.11° N | 12/10 22:08 | 1027.5±177 | 721.3 | 638 | Night |
| Russia | GRES | 57.11° N | 2/8 08:29 | 1109±169 | 721.3 | 638 | Day |
| Russia | GRES | 57.11° N | 7/1 22:08 | 724.5±115 | 721.3 | 638 | Night |
| America | Scherer | 33.06° N | 5/3 02:37 | 478±72 | 267.4 | 607.5 | Night |
| Poland | Belchatow | 51.26° N | 5/8 19:18 | 771±134 | 867.5 | 925 | Day |
| South Africa | Medupi | 23.71° S | 7/24 07:21 | 598±98 | 516.6 | 515.3 | Day |
| South Africa | Matimba | 23.60° S | 7/24 07:21 | 708±118 | 617 | 664.6 | Day |
| Russia | CHP-1 | 69.33° N | 6/14 21:07 | 109.7±18 | 83 | -- | Night |
| Russia | CHP-3 | 69.32° N | 6/14 21:07 | 57.1±12 | 44.1 | -- | Night |



| Russia | GRES-2 | 61.28° N | 7/24 21:36 | 1287.3±143 | 1001.1 | -- | Night |
| Korean | Taean | 36.90° N | 6/03 04:44 | 991.5±73 | 1022.2 | 900.5 | Day |
| Korean | Daesan | 36.99° N | 6/03 04:44 | 30.4±3.4 | 23.9 | -- | Day |




**Figure 8: Comparison of power plant emissions predicted by Gaussian plume model with Climate TRACE statistics, the solid black line represents the 1:1 line, and the dashed line indicates the linear fitting line.**

Overall, the CO₂ emissions predicted by the Gaussian plume model are generally higher than those in the emissions inventory. This is because some of the observations were made during the winter and summer in the Northern Hemisphere when residents' demand for electricity, such as air conditioning, increases, prompting power plants to raise their generation capacity. Comparing nighttime CO₂ emissions with daytime observations shows that emissions from some power plants are lower at night, primarily due to reduced electricity demand at night (Waite et al., 2017), and some power stations adjust power generation in real-time to avoid power storage saturation. However, for some plants, nighttime emissions exceed those in the inventory, possibly because when the load on the equipment is below its optimal level, the plant's overall efficiency decreases, leading to incomplete fuel combustion. Additionally, frequent start-stop operations during low-load conditions may cause unstable combustion, further increasing emissions (Hendriks, 2012). Overall, the predicted emissions are slightly higher than those reported by Carbon Brief and Climate TRACE, mainly because conventional power plants undergo annual shutdowns

for equipment inspections, resulting in lower annual averages compared to instantaneous emissions. Moreover, by utilizing
the high spatial resolution of the DQ-1 satellite, it is possible to monitor low $CO_2$-emitting power plants (F < 100 kg s$^{-1}$), with
results fitting well with the inventory data.

## 4 Conclusions

This study utilized the IPDA lidar aboard the DQ-1 satellite to monitor emissions from localized strong point sources and, for
the first time, observed the diurnal variation of $CO_2$ emissions from a high-latitude power plant, effectively covering areas that
passive remote sensing satellites fail to monitor. The two-dimensional Gaussian plume model was optimized in terms of plume
direction and atmospheric stability and applied to $XCO_2$ observation results. Validation and comparison results indicate that
the improved Gaussian plume model has a strong correlation with the emissions inventory, with a correlation coefficient of
0.97. The average relative random error of the predicted results is 15.11 %, which is lower than that of the EMI-GATE model,
due to different parameter selections in the Gaussian plume model, thus reducing the random error. The main factors affecting
estimation errors are the uncertainty in the atmospheric wind field (26.7 % of total error), uncertainty in atmospheric stability
(25.1 %), and uncertainty in background field calculations (40.7 %). Establishing automatic weather stations around the power
plant for real-time monitoring of atmospheric radiation and surface wind speed could reduce errors caused by uncertainties in
atmospheric stability. Overall, power plant $CO_2$ emissions were largely consistent with local electricity consumption patterns,
with most plants emitting less at night than during the day, and with higher emissions in winter and summer compared to
spring and autumn. This research provides a new approach for global carbon accounting. In 2025, China will launch the DQ-
2 satellite, equipped with the same IPDA lidar for carbon dioxide observation. As satellite density increases, global coverage
of emissions detection data will significantly improve.

**Data availability.**

ERA-5: https://cds.climate.copernicus.eu/#!/home.
MERRA-2: https://gmao.gsfc.nasa.gov/reanalysis/MERRA-2/
Carbon brief: https://www.carbonbrief.org/mapped-worlds-coal-power-plants/
Climate TRACE: https://climatetrace.org/explore/electricity-generation-co2e100-2022

**Author contributions.**

XZ, HY and LB designed and directed the study. XZ and ZF contributed to data analysis and wrote the first draft of this paper.
BC, LZ, SL, ZW and WC collected data. JL, XL and WX contributed the data interpretation and review of the paper.



**Competing interests.**

The authors declared that they have no conflict of interest.

**Disclaimer.**

Publisher's note: Copernicus Publications remains neutral with regard to jurisdictional claims made in the text, published maps, 365 institutional affiliations, or any other geographical representation in this paper. While Copernicus Publications makes every effort to include appropriate place names, the final responsibility lies with the authors.

**Acknowledgements.**

We acknowledge Shanghai Institute of Optics and Fine Mechanics, Chinese Academy of Sciences and National Satellite Meteorological Center for providing the DQ-1 data. We express our gratitude to the teams that produce and maintain the high- 370 quality meteorological data used in this study from ERA-5, Climate TRACE, MERRA-2, Carbon Brief.

**Financial support**

This research was funded by the National Key Research and Development Program of China (Grant No. 2020YFA0607501), National Natural Science Foundation of China (Grant No. 42175145), International Partnership Program of Chinese Academy of Sciences (18123KYSB20210013), Shanghai Science and Technology Innovation Action Plan (22dz208700).

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
