# Peer review of "Estimation of diurnal emissions of CO2 from thermal power plants using spaceborne IPDA lidar"

_EGUsphere, 2024_

## Referee Comment (RC1)

**Review of the manuscript:**

**Estimation of diurnal emissions of CO2 from thermal power plants using spaceborne IPDA lidar**

Authors: Xuanye Zhang, Hailong Yang, Lingbing Bu, Zengchang Fan, Wei Xiao, Binglong Chen, Lu Zhang, Sihan Liu, Zhongting Wang, Jiqiao Liu, Weibiao Chen and Xuhui Lee

**Synopsis:**

This paper presents a comprehensive analysis of XCO2 observations obtained in the vicinity of coal-fired power plants using the ACDL spaceborne instruments on board the DQ-1 satellite. Leveraging the integrated differential absorption (IPDA) lidar technique, the work utilizes these data to derive emission fluxes and compare them against existing inventories.

The IPDA method offers a significant advantage over passive remote sensing technologies by enabling measurements at night and in high-latitude regions, where traditional methods are often limited or ineffective. By employing a Gaussian plume model to interpret the observed XCO2 signatures, this study aims to infer CO2 emissions from coal-fired power plants.

**General Review:**

In my opinion, this contribution provides a substantial step forward to infer CO2 emission from satellite-borne instruments. As the first of its kind CO2 lidar in space, the work provides new data and thus is significant for the scope of Atmospheric Chemistry and Physics.

The scientific approach and methodologies employed are clearly presented, and the results are discussed in a largely satisfactory manner. However, to strengthen the paper's credibility, I recommend adding references to relevant existing studies. The results themselves are presented concisely and effectively, with conclusions drawn by the authors that are well-supported.

The scientific approach and the methods applied are well presented. The results are mostly discussed appropriately, however, several references to related work need to be added. The results are presented in a clear and concise way. The authors developed their conclusion well. Some of the figures should be improved for the sake of readability and, language-wise, some expressions need improvement.

To estimate CO2 emissions from power plants, the authors apply a Gaussian-shaped plume model derived from theoretical considerations.
In order to estimate the CO2 emissions of power plants the authors fit the results to a Gaussian-shaped plume derived from model. It is suggested that the authors provide justification for choosing this approach over alternative methods, such as budgeting (Riemann sum) approaches e.g. https://doi.org/10.1364/AO.56.005182 .

The most significant deficiency of the paper is its failure to discuss potential implications of turbulence on CO2 emission estimates. Turbulence can significantly impact the validity of applying a Gaussian plume approximation and may lead to biases in deriving emission rates, especially considering the expected differences between night-time and daylight measurements. This topic requires a more detailed discussion, given its complexity.

This is more detailed in the specific review below.

**Specific Review:**

**Abstract and Introduction:**

- The authors' conclusion in the abstract that their method is adequate for monitoring diurnal variations in power plant emissions strikes a note of caution, given that the satellite only overpasses the sources once per night and once per day, which does not provide better time resolution than what the method can account for.

- The authors provide a comprehensive review of studies utilizing passive remote sensing techniques. Notably, they accurately highlight the benefits of active over passive remote sensing methods. Nevertheless, it would be beneficial for them to extend this thoroughness to lidar-related references, which appear to be incomplete.
  At least the following publications need to be cited:

  - Ehret et al.: https://doi.org/10.1007/s00340-007-2892-3

  - Menzies et al., https://doi.org/10.1175/JTECH-D-13-00128.1

  - Wolff et al.: https://doi.org/10.5194/amt-14-2717-2021
    This paper does not only present airborne IPDA measurement of CO2 plumes from power plants, but also discusses the mass balance and Gaussian plume approximation using lidar cross sections. Moreover, it discusses turbulence and its impact on IPDA measurements. Thus, it is of high relevance to this paper.

  - Kiemle et al.: https://doi.org/10.3390/rs9111137

Even for results for passive remote sensing (eg. w.r.t EM27/SUN) a few important publications are missing and should be cited:

  - Luther et. al: https://doi.org/10.5194/amt-12-5217-2019
  - Ye et al.: https://doi.org/10.1029/2019JD030528
  - Wu et al.: https://doi.org/10.5194/gmd-11-4843-2018

Concerning plume shapes and results from different models, Brunner et al. need to be cited.
https://doi.org/10.5194/acp-23-2699-2023

- The number of power plants/ observations seems to be inconsistent within the abstract. It becomes not so clear how many individual power plants have been observed (10?), how many of those have been overflown more than once, how many overflights were considered in total (38?) and how many overflights took place at night (15?) and day (23?). Please clarify. Possibly, table 2 can be improved with this regard. It would also be helpful to know about the percentage of "good" overflights versus all overflights.

- Be more specific in what is improved versus the Han et al. paper.

**Data and Methodology**

Line 113:   Explain why water vapour column information is needed.

Para 2.1.3:   Can you comment on the accuracy of the Climate Trace Data and specifically about the emissions of those power plants considered in this study.

Line 125:   Bovensmann et al. is certainly not the original reference for the Gaussian plume approximation. The authors should probably look for a better suited reference.

Line 135:   "gravitational acceleration" (rather than acceleration due to …)

Figure 1:   Please improve the graphs such that the labelling of axes can be read.

Line 148:   Please describe why you use a 30-km distance as the limit.

Line 153:   Please describe why you use a 25° angle as the limit. It would also be interesting to know how many overflights had to be discarded due to wind directions outside of this criterion, i.e. the percentage of "good" overpasses versus all overpasses.

Line 159:   Please describe how cloud cover and solar altitude are considered in the study of atmospheric stability. What about solar altitude and night-time measurements?

Line 160:   Over which distance smoothing was applied. In this context, the authors should be more precise. At some point the precision of the measurements (for the applied averaging interval) should be given.

Para 2.2.:   The authors should better distinguish between stack height and plume heights. To do so, a paper by Brunner et al. is considered to be important and must be cited: https://doi.org/10.5194/acp-19-4541-2019

**Results and Discussion**

Line 186:   Here, a radius of 50 km is given. This is inconsistent with the 30-km criterion given above. Correct or explain!

Line 188:   Please be quantitative in specifying the filter criteria. For example, what minimum optical depth of "thick" clouds leads to exclusion?

Line 199:   Pleas include (e.g. in Table 1 or 2) for which power plants the stack height is known to the authors. Moreover, the stack height is not uncertain, it is just not known to the authors.

Line 200:   What is meant by: "OCO-2 satellites have no valid Gaussian plume data?

Table 2:   Please provide also the year of the respective overpass.

Figure 2: The source of the maps should be given. It would be nice to provide a map insert providing a better information about where the GRES power plant is located. Please increase the axis labels on the left figure.

Figure 3: Please increase the figures and labels for better readability. It might be more instructive to show the overpasses in the same map cutout and scale. Is there a significance in the size of the wind arrows? I suggest to provide approximate wind speed information and local time of overpass. Probably is makes no sense to apply the same scale to all graphs due to different background values, but at least the same spread (2 ppm?) should be applied to all figures.

Line 216: How is the lift height derived from stack height and atmospheric stability. Does the temperature of the gas play a role?

Line 253: Provide a number for the linearly varying background.

Figure 7: Climate Trace (not Change?); The reader may think that this is continuous data, but the lines consist of 7? individual daytime and 9? nighttime observations. Please plot the data points!

Line 298: Which power plants provide stack heights? Include this information in one of the Tables.

Line 300: The authors must comment on the single pulse-pair precision that is used to fit the Gaussian plume. On short spatial scales, this information is more important than the accuracy of the IPDA measurement since the background is subtracted.

Table 1/2: Swap the order of the tables. Table 1 should contain the basic information about the power plants and Table 2 the uncertainties.

Figure 8: See comment above about the error of the data base.

Line 325: Without robust evidence on the diurnal (day-night) and seasonal (winter-summer) variations in power plant emissions, any claims about these differences are speculative and should be clearly labeled as such.

Line 333: Quantify the higher emission rates vs. Carbon Brief and Climate TRACE.

**Conclusions:**

In the conclusions, I would like to see suggestions for potential improvements in emission estimate accuracy, such as utilizing more advanced models or those with higher spatial resolution. For instance, the authors briefly mention WRF (Weather Research and Forecasting) model in the introduction. However, it would be helpful to provide a more detailed discussion on what enhancements can be expected from using this model instead of a Gaussian plume model, as well as any challenges that might arise.

**Data availability:**

Please give a hint about the availability of ACDL data, since this is the major data source used in this work.

**Summary:**

To summarize, the manuscript presents valuable findings based on new data, but also exhibits some shortcomings, including missing citations and a critical oversight regarding the impact of turbulence on the results. Notably, this omission is particularly concerning in light of potential differences between daytime and nighttime measurements. In order to ensure the manuscript's credibility and thoroughness, I believe it is essential to address these issues before publication.

---

## Author Response (AR1)

**Response to RC1:**

Authors: Xuanye Zhang, Hailong Yang, Lingbing Bu, Zengchang Fan, Wei Xiao, Binglong Chen, Lu Zhang, Sihan Liu, Zhongting Wang, Jiqiao Liu, Weibiao Chen and Xuhui Lee

**Synopsis:**

This paper presents a comprehensive analysis of XCO2 observations obtained in the vicinity of coal-fired power plants using the ACDL spaceborne instruments on board the DQ-1 satellite. Leveraging the integrated differential absorption (IPDA) lidar technique, the work utilizes these data to derive emission fluxes and compare them against existing inventories.

The IPDA method offers a significant advantage over passive remote sensing technologies by enabling measurements at night and in high-latitude regions, where traditional methods are often limited or ineffective. By employing a Gaussian plume model to interpret the observed XCO2 signatures, this study aims to infer CO2 emissions from coal-fired power plants.

**Answer: We greatly appreciate your valuable time for reviewing our research paper and providing suggestions.** *(The blue text is in response to your comments, and the green text is for specific modifications in the paper. We also highlight revisions in the manuscript.)*

**General Review:**

In my opinion, this contribution provides a substantial step forward to infer CO2 emission from satellite-borne instruments. As the first of its kind CO2 lidar in space, the work provides new data and thus is significant for the scope of Atmospheric Chemistry and Physics.

The scientific approach and methodologies employed are clearly presented, and the results are discussed in a largely satisfactory manner. However, to strengthen the paper's credibility, I recommend adding references to relevant existing studies. The results themselves are presented concisely and effectively, with conclusions drawn by the authors that are well-supported.

The scientific approach and the methods applied are well presented. The results are mostly discussed appropriately, however, several references to related work need to be added. The results are presented in a clear and concise way. The authors developed their conclusion well. Some of the figures should be improved for the sake of readability and, language-wise, some expressions need improvement.

To estimate CO2 emissions from power plants, the authors apply a Gaussian-shaped plume model derived from theoretical considerations.

In order to estimate the CO2 emissions of power plants the authors fit the results to a Gaussian-shaped plume derived from model. It is suggested that the authors provide justification for choosing this approach over alternative methods, such as budgeting (Riemann sum) approaches e.g. https://doi.org/10.1364/AO.56.005182 .

The most significant deficiency of the paper is its failure to discuss potential implications of turbulence on CO2 emission estimates. Turbulence can significantly impact the validity of applying a Gaussian plume approximation and may lead to biases in deriving emission rates, especially considering the expected differences between night-time and daylight measurements. This topic requires a more detailed discussion, given its complexity.

This is more detailed in the specific review below.

Answer:

We sincerely appreciate the reviewer's valuable comments and suggestions. We have carefully revised the manuscript and figures in accordance with the specific recommendations provided. **Regarding the first comment** on "*It is suggested that the authors provide justification for choosing this approach over alternative methods, such as budgeting (Riemann sum) approaches e.g.*" : While budgeting approaches (e.g., Riemann sum) do not require assumptions about the specific morphology of emission plumes (e.g., Gaussian distribution) and are more robust for complex or irregular plume distributions, the primary rationale for adopting Gaussian fitting in this study lies in our focus on detecting emissions from power plants in **nighttime and high-latitude** regions. For high-latitude plants, field observations reveal that most are located inland in areas with minimal interference from other anthropogenic emissions. By rigorously filtering cases based on wind direction consistency (deviation <25° from the plume axis) and cloud-free conditions, the atmospheric wind fields for the corresponding cases are relatively stable, resulting in emission plumes that closely approximate Gaussian distributions. The Gaussian fitting method allows us to determine the plume shape by incorporating atmospheric stability parameters and wind field variables, thereby enhancing model prediction accuracy in point-to-point fitting. Furthermore, this approach enables systematic uncertainty analysis for different error sources (e.g., wind field errors, background $XCO_2$ variability). We have revised in Line 58 : "Compared to the budgeting approach for estimating point source emissions (Amediek et al., 2017), the Gaussian plume model (GPM) is highly influenced by the precision of atmospheric background driving fields (Nassar et al., 2017; Brunner et al., 2023), it is not constrained by the spatial resolution of the model, and is more stable and effective in modelling point source dispersion if limited by the background wind field (Toja-Silva et al., 2017; Schwandner et al., 2017)."

**Regarding the second comment** "*The most significant deficiency of the paper is its failure to discuss potential implications of turbulence on $CO_2$ emission estimates*" we clarify that turbulence impacts the Gaussian plume model (GPM) in two primary aspects. Firstly, turbulence induces rapid changes in atmospheric wind fields, leading to non-Gaussian $XCO_2$ enhancement patterns in satellite observations. To address this, we excluded such cases through rigorous data filtering criteria. In Section 3, we retained only cases where the angle between the plume axis and wind direction was <25° . When the deviation is <25° , we attribute the discrepancy primarily to interpolation errors in ERA5 wind fields (the rationale for the 25° threshold will be elaborated in subsequent responses). For deviations ≥25°, observed $XCO_2$ profiles deviate significantly from Gaussian shapes, likely due to turbulence effects, rendering Gaussian fitting unreliable for emission quantification; thus, these cases were discarded. Secondly, turbulence

affects the calculation of the horizontal dispersion coefficient $\sigma_y$ (parameter a in Function 1). In this study, atmospheric stability was determined using Gordon's implementation of the Pasquill-Gifford scheme (Tables S1/S2/S3). We computed net radiation indices from ERA5 total/low cloud cover and solar elevation angles, then combined these with surface wind speeds to linearly interpolate the initial atmospheric stability parameter $a_1$. The standard deviation of surface wind speeds was used to define a 1 $\sigma$ uncertainty range for $a_1$, within which both a and $CO_2$ emission rates were co-optimized to determine the optimal $a_2$. The Pasquill classification effectively quantifies molecular diffusion: under strong daytime solar radiation, intensified turbulence enhances mixing, resulting in larger $\sigma_y$ values, lower Gaussian peaks, and flatter plume distributions. However, our results indicate that daytime turbulence introduces errors in $\sigma_y$ estimation via the Pasquill method, leading to higher uncertainties in emission quantification. Conversely, nighttime atmospheric stability yields better agreement between simulated and observed plume distributions, with reduced errors from atmospheric instability. In the article we have analysed the errors due to uncertainty in atmospheric instability, and we have analysed the diurnal differences in the errors due to atmospheric instability separately in the discussion of the results Line 387 : "The results show that during the daytime, the error in the surface wind field is higher due to turbulence, which can cause some invalid observations or increase the error caused by atmospheric instability to the model.", the errors due to atmospheric instability have been further explained in Line 371: "Unlike prior studies, this research explicitly accounts for atmospheric instability uncertainty. Surface wind speed uncertainty, influenced by boundary layer turbulence and other factors, is significantly higher during daytime. Our analysis of 28 cases reveals that the optimized atmospheric instability parameter a shows average deviations of 19.5% from its prior value in daytime versus 15.8% at night. The results indicate that, under the assumption that plume dispersion aligns with the Gaussian plume model, ERA-5 surface wind speeds exhibit higher accuracy at night. However, daytime turbulence introduces small-scale wind field errors, which further amplify uncertainties in atmospheric instability."

$$\sigma_y(x) = a \cdot (x/1000)^{0.894} \tag{1}$$

Table S1

| Total cloud cover/low cloud cover (0-1) | Net Radiation Index | | | | |
|---|---|---|---|---|---|
| | nightime | solar altitude angle | | | |
| | | $\leq 15°$ | $15°-35°$ | $35°-65°$ | $> 65°$ |
| $\leq 0.4/\leq 0.4$ | -2 | -1 | +1 | +2 | +3 |
| 0.5-0.7/$\leq 0.4$ | -1 | 0 | +1 | +2 | +3 |
| $\geq 0.8/\leq 0.4$ | -1 | 0 | 0 | +1 | +1 |
| $\geq 0.5/0.5-0.7$ | 0 | 0 | 0 | 0 | +1 |
| $\geq 0.8/\geq 0.8$ | 0 | 0 | 0 | 0 | 0 |

Table S2

| Surface wind speed (m/s) | Net Radiation Index(Table S1) | | | | | |
|---|---|---|---|---|---|---|
| | 3 | 2 | 1 | 0 | -1 | -2 |
| <2 | A | A-B | B | D | (E) | (F) |
| 2-3 | A-B | B | C | D | E | F |
| 3-5 | B | B-C | C | D | D | E |
| 5-6 | C | C-D | D | D | D | D |
| >6 | C | D | D | D | D | D |

Table S3

| stability parameter | A | B | C | D | E | F |
|---|---|---|---|---|---|---|
| Parameter a | 213 | 184.5 | 150 | 130 | 104 | 77.6 |

**Specific Review:**

**Abstract and Introduction:**

1. The authors' conclusion in the abstract that their method is adequate for monitoring diurnal variations in power plant emissions strikes a note of caution, given that the satellite only overpasses the sources once per night and once per day, which does not provide better time resolution than what the method can account for.

Answer:Thanks, we modify diurnal variation to diurnal difference. We've also revised in Line 27: "This paper also analyses the diurnal differences in CO2 emissions from power plants and finds emissions fluctuations directly correlated with regional electricity demand dynamics. This method is very effective for monitoring emissions from strong point sources such as power plants."

2. The authors provide a comprehensive review of studies utilizing passive remote sensing techniques. Notably, they accurately highlight the benefits of active over passive remote sensing methods. Nevertheless, it would be beneficial for them to extend this thoroughness to lidar-related references, which appear to be incomplete.

At least the following publications need to be cited:

Ehret et al.: https://doi.org/10.1007/s00340-007-2892-3

Menzies et al., https://doi.org/10.1175/JTECH-D-13-00128.1

Wolff et al.: https://doi.org/10.5194/amt-14-2717-2021

Answer: Thanks for the suggestion that these active remote sensing references should be mentioned, we have added the following to line 75: "Spaceborne active remote sensing of $CO_2$ primarily employs the integrated path differential absorption (IPDA) principle (Ehret et al., 2008), enabling nighttime and high-latitude observations. Kiemle et al. discussed the ability to monitor $CO_2$ emissions using spaceborne lidar in combination with the plume model and a mass budget approach (Kiemle et al., 2017). Recent studies have demonstrated the feasibility of laser-based detection techniques for $CO_2$ emission monitoring (Menzies et al., 2022; Wolff et al., 2024)."

3. This paper does not only present airborne IPDA measurement of CO2 plumes from power plants, but also discusses the mass balance and Gaussian plume approximation using lidar cross sections. Moreover, it discusses turbulence and its impact on IPDA measurements. Thus, it is of high relevance to this paper.

Kiemle et al.: https://doi.org/10.3390/rs9111137

Answer: Thanks, we have added the contribution of Kiemle et al. in line 76: "Kiemle et al. discussed the ability to monitor $CO_2$ emissions using spaceborne lidar in combination with the plume model and a mass budget approach (Kiemle et al., 2017)."

4. Even for results for passive remote sensing (eg. w.r.t EM27/SUN) a few important publications are missing and should be cited:

− Luther et. al: https://doi.org/10.5194/amt-12-5217-2019

− Ye et al.: https://doi.org/10.1029/2019JD030528

− Wu et al.: https://doi.org/10.5194/gmd-11-4843-2018

Answer: Thanks, we have added references to these literature in the citation and revised in the paper. Line 46: "In recent years, some studies have used "top-down" approaches, such as combining satellite observation data with WRF-STILT or WRF-Chem models to quantify urban greenhouse gas emissions (Turner et al., 2020; Pillai et al., 2012; Hu et al., 2022; Wu et al., 2018; Ye et al., 2020)." Line 54: "Ground-based equipment, such as portable Fourier transform spectrometers (EM27/SUN), combined with the Gaussian plume model and the cross-sectional flux method, has also been used to measure ground-level $CO_2$ concentrations for specific power plants and urban areas (Ohyama et al., 2021; Luther et al., 2019)."

5. Concerning plume shapes and results from different models, Brunner et al. need to be cited. https://doi.org/10.5194/acp-23-2699-2023

Answer: Thanks, we have cited this reference and illustrated the need for plume models to incorporate high-precision background wind fields,in Line : "Compared to the budgeting approach for estimating point source emissions (Amediek et al., 2017), the Gaussian plume model (GPM) is highly influenced by the precision of atmospheric background driving fields (Nassar et al., 2017; Brunner et al., 2023), it is not constrained by the spatial resolution of the model, and is more stable and effective in modelling point source dispersion if limited by the

background wind field (Toja-Silva et al., 2017; Schwandner et al., 2017)."

6. The number of power plants/ observations seems to be inconsistent within the abstract. It becomes not so clear how many individual power plants have been observed (10?), how many of those have been overflown more than once, how many overflights were considered in total (38?) and how many overflights took place at night (15?) and day (23?). Please clarify. Possibly, table 2 can be improved with this regard. It would also be helpful to know about the percentage of "good" overflights versus all overflights.

Answer:In total, we analysed 10 power plants with a total of 28 observations. For the GRES plant, we analyzed a total of 7 daytime and 11 nighttime results in order to analyze the diurnal and seasonal differences in $CO_2$ emissions, and for the remaining 9 plants, the main role in this study was to compare with the emission inventories and to validate the correctness of the model, so only one valid observation was selected. In response to the reviewer's question, the 15 times are the number of nighttime results and the 23 times are the number of high-latitude observations, not the number of daytime results. In response to the reviewer's second query, a total of 97 overpasses were obtained around several power plants, resulting in a total of 28 'Good' overpasses used in this paper.In Line 205: "A total of 97 satellite overpasses within 50-km circular regions centered on plant stacks were retrieved…...After filtering, 34 % of the overpasses were discarded due to excessive differences in wind direction, and a total of 28 overpasses were finally selected, including 15 nighttime cases and 3 cases where the power plant was located above 60° N."

7. Be more specific in what is improved versus the Han et al. paper.

Answer:The key distinction between the Gaussian plume model employed in this study and the EMI-GATE model used by Han et al. lies in the calculation methods for atmospheric instability parameters, background $XCO_2$ column concentrations, wind speeds, and wind directions. Our approach involves varying these parameters within their respective error bounds based on raw observational data, with each parameter adjusted independently to maximize the interpretability of results. In contrast, Han et al. utilized a Genetic Algorithm and Trust-rEgion (GATE) technique to simultaneously optimize multiple variables including emission rates, wind speeds, and atmospheric instability parameters. While this enhances model fitting performance, it compromises the interpretability of the results compared to our method. Furthermore, this study used different atmospheric instability calculation method and explicitly quantifies uncertainties arising from atmospheric instability, which were not systematically analysed in Han et al.'s methodology. In addition, in terms of the content of the study, two years of observations were obtained for a single power plant and the temporal differences in emissions were analysed. We have revised in Line 85: "The main differences between the Gaussian plume model used in this study and the EMI-GATE model used by Han et al. are the methods used to calculate the Gaussian plume model parameters such as the atmospheric instability parameters and the wind field, as well as the fact that we additionally quantify the uncertainty due to atmospheric instability."

**Data and Methodology**

8. Line 113: Explain why water vapour column information is needed.

Answer:Thanks, in order to convert $CO_2$ column increment to $XCO_2$, surface pressure and water vapour

column content are required in Equation (2).

$$\Delta XCO_2(x,y) = \Delta Q(x,y) \cdot \frac{M_{air}}{M_{CO_2}} \cdot \frac{g}{P_{surf} - w \cdot g} \cdot 1000 \qquad (2)$$

We have revised in Line 124: "Additionally, the conversion of emissions into $XCO_2$ enhancements requires surface pressure and water vapor column content data, which were derived from ERA5-Land datasets."

9. Para 2.1.3: Can you comment on the accuracy of the Climate Trace Data and specifically about the emissions of those power plants considered in this study.

Answer: Since accurate emission data are unavailable, we conducted a cross-sectional comparison between inventories. We selected top 30 high-emission power plants and compared results from Climate TRACE with those from the Carbon Brief inventory, yielding an average relative deviation of 9.2%. Climate TRACE integrates machine learning with observational data, providing broader coverage of power plants—particularly smaller emitters. Among the 10 plants analyzed in this study, 6 have data in both inventories. For the Scherer power plant, emissions differed by a factor of two between the two inventories, while the average deviation for the remaining plants was 7 %. The primary rationale for selecting Climate TRACE data lies in its machine learning-derived monthly-resolution emission estimates, which facilitate temporal alignment with our results. We explained in line 133. "We also compared the emissions of Climate TRACE's top 30 power plants with Carbon Brief, which had an average deviation of 9.2 %, and we considered their results to be reliable."

10. Line 125: Bovensmann et al. is certainly not the original reference for the Gaussian plume approximation.The authors should probably look for a better suited reference.

Answer: Thanks, we have revised it to: *Pasquill, Frank, and Frank Barry Smith. Atmospheric diffusion. Vol. 437. Chichester: Ellis Horwood, 1983.*

11. Line 135: "gravitational acceleration" (rather than acceleration due to …)

Answer: Revised

12. Figure 1: Please improve the graphs such that the labelling of axes can be read.

Answer: Revised

13. Line 148: Please describe why you use a 30-km distance as the limit.

Answer: 30-km was incorrect and has been revised to 50 km. For a power plant with an emission rate of 1000 Kg/s, $\Delta XCO_2$ is less than 1 ppm at 50 km downwind of the plume at a wind speed of 10 m/s, which is less than the uncertainty of the $XCO_2$ observed by the DQ-1 satellite, and we believe that the results must be inaccurate at a distance far away from this distance, and therefore this condition was added during the screening of the data. It is also explained in the Line 160: "For a power plant with an emission rate of 1000 kg s-1, the downwind $XCO_2$ enhancement at 50 km under 10 m s-1 winds is <1 ppm, which is less than the uncertainty of $XCO_2$ observed by the DQ-1, indicating low reliability in

distant plume detection. Therefore, to improve the model's fit with satellite observation results, the selected DQ-1 orbital data require that the downwind direction of the point source intersects with the satellite footprint and that the distance between the $XCO_2$ enhancement location and the point source is less than 50 km."

14. Line 153:  Please describe why you use a 25° angle as the limit. It would also be interesting to know how many overflights had to be discarded due to wind directions outside of this criterion, i.e. the percentage of "good" overpasses versus all overpasses.

Answer:  Thank you for your comments!

1)We consider that deviations of wind direction from the plume centre pointing of less than 25% are likely to be due to errors in the meteorological data, while data greater than 25% are likely to be due to large effects of atmospheric turbulence (Panofsky et al., 1984), which are not suitable for fitting with a Gaussian model.We revised in Line 206: "For 10 typical coal-fired power plants, 47 satellite overpasses intersecting the downwind direction were identified within the power plant area. Strict data filtering standards were applied when using the Gaussian plume model, requiring the absence of thick clouds (DQ-1 measured elevation discrepancies from DEM <100 m) and ensuring that the deviation of the wind direction from the plume spreading direction at the point source was less than 25°. We consider that deviations of wind direction from the plume axis of less than 25 % are mainly attributable to meteorological data uncertainties, while larger deviations ($\geqslant$25 %) may be due to atmospheric turbulence effects (Panofsky et al. 1984), when Gaussian plume modelling is not appropriate."

2)  For the second question raised by the reviewer, there are a total of 97 overpasses in the study area, 47 of which have their tracks intersecting the downwind direction, and after screening the wind field and clouds, there are only 28 "good" overpasses, with a percentage of good results of about 30 %. We have revised in Line 212: "After filtering, 34 % of the overpasses were discarded due to excessive differences in wind direction, and a total of 28 overpasses were finally selected, including 15 nighttime cases and 3 cases where the power plant was located above 60° N."

15. Line 159:  Please describe how cloud cover and solar altitude are considered in the study of atmospheric stability. What about solar altitude and night-time measurements?

Answer: Thank you for your comments, in this study, atmospheric stability was determined using Gordon's implementation of the Pasquill-Gifford scheme (Tables S1/S2/S3). Solar altitude can be calculated by combining latitude, time and Equation 3. There are also corresponding results in Table S1 for the night time case.

$$\delta = \begin{bmatrix} 0.006918 - 0.399912 \cos \theta_0 + 0.070257 \sin \theta_0 - 0.006758 \cos 2\theta_0 \\ +0.000907 \sin 2\theta_0 - 0.002697 \cos 3\theta_0 + 0.001480 \sin 3\theta_0 \end{bmatrix} \times \frac{180°}{\pi} \tag{3}$$

We have also revised in Line 174: "In this study, empirical interpolation of atmospheric stability parameters was implemented by accounting for surface wind speed, cloud coverage, and solar elevation angle, the latter calculated from latitude and time of observation (Nassar et al., 2021)."

16. Line 160:  Over which distance smoothing was applied. In this context, the authors should be more precise. At some point the precision of the measurements (for the applied averaging interval)

should be given.

Answer: In this study, a moving average over a distance of 10 km is used so that the $XCO_2$ uncertainty is less than 1 ppm, which corresponds to the pulse signals received by the satellite, and a moving average is performed for 30 pulses. We have revised in Line 179: "A 10-km moving averaging was applied to DQ-1 data, reducing the uncertainty of $XCO_2$ to below 1 ppm, which facilitated the detection of enhanced $XCO_2$ signals (Zhang et al., 2023)."

17. Para 2.2.: The authors should better distinguish between stack height and plume heights. To do so, a paper by Brunner et al. is considered to be important and must be cited: https://doi.org/10.5194/acp-19- 4541-2019

Answer:We have explained in the paragraph 2 that plume height = stack height + plume lift height, and add the reference. In Line 195: "The plume height is equal to the stack height plus the plume lift height (Brunner et al., 2019), and if there is no information on the stack height for a specific power plant, the default stack height is 240 m, and given the uncertainty in the plume lift height, the standard deviation of the emissions for lift heights of 160 m, 200 m, 240 m, 280 m, and 320 m is used to estimate the uncertainty."

**Results and Discussion**

18. Line 186: Here, a radius of 50 km is given. This is inconsistent with the 30-km criterion given above. Correct or explain!

Answer: Thanks for the correction. 50km is correct and has been corrected in the paper

19. Line 188: Please be quantitative in specifying the filter criteria. For example, what minimum optical depth of "thick" clouds leads to exclusion?

Answer:The condition for IPDA lidar to recognise whether there is cloud interference or not is not differentiated by optical thickness, DQ-1 also emits 1064 nm laser to measure the ground elevation, when there is a thick cloud, the laser elevation is the height of the top of the cloud, we compare the laser elevation with the Digital Elevation Model, the signal with a difference of more than 100m we consider it as a thick cloud interference (Gao et al., 2023), and we assume that this signal is invalid. We have also revised in Line 208: "Strict data filtering standards were applied when using the Gaussian plume model, requiring the absence of thick clouds (DQ-1 measured elevation discrepancies from DEM <100 m) and ensuring that the deviation of the wind direction from the plume spreading direction at the point source was less than 25°."

*Xuejie Gao, Jiqiao Liu, Chuncan Fan, Cheng Chen, Juxin Yang, Shiguang Li, Yuan Xie, Xiaopeng Zhu, Weibiao Chen. Carbon Dioxide Column Concentration Measurement Based on Cloud Echo Signal of 1.57 μm IPDA Lidar[J]. Chinese Journal of Lasers, 2023, 50(23): 2310001.*

20. Line 199: Pleas include (e.g. in Table 1 or 2) for which power plants the stack height is known to

the authors. Moreover, the stack height is not uncertain, it is just not known to the authors.

Answer:We have searched and found the stack heights of four larger power plants and have added this column to the table. The main factor affecting the uncertainty in the calculation of the plume height is the uncertainty in the plume lift height, we have revised in Line 195: "The plume height is equal to the stack height plus the plume lift height (Brunner et al., 2019), and if there is no information on the stack height for a specific power plant, the default stack height is 240 m, and given the uncertainty in the plume lift height, the standard deviation of the emissions for lift heights of 160 m, 200 m, 240 m, 280 m, and 320 m is used to estimate the uncertainty."

**Table 1 Information on different power plants and the comparison of model predictions with emission inventories**

| Country | Station | Stack Height (m) | Latitude | UTC Time | Model Result (kg s$^{-1}$) | Climate TRACE (kg s$^{-1}$) | Carbon Brief (kg s$^{-1}$) | Day or Night |
|---|---|---|---|---|---|---|---|---|
| Russia | GRES | 330 | 57.11° N | 2022/8/17 22:12 | 806.0±108 | 721.3 | 638 | Night |
| Russia | GRES | 330 | 57.11° N | 2023/5/22 08:35 | 876.2 ±153 | 721.3 | 638 | Day |
| Russia | GRES | 330 | 57.11° N | 2023/11/05 22:10 | 988.6±161 | 721.3 | 638 | Night |
| Russia | GRES | 330 | 57.11° N | 2023/12/10 22:08 | 1027.5±177 | 721.3 | 638 | Night |
| Russia | GRES | 330 | 57.11° N | 2024/2/8 08:29 | 1109±169 | 721.3 | 638 | Day |
| Russia | GRES | 330 | 57.11° N | 2024/7/1 22:08 | 724.5±115 | 721.3 | 638 | Night |
| America | Scherer | 305 | 33.06° N | 2022/5/3 02:37 | 478±72 | 267.4 | 607.5 | Night |
| Poland | Belchatow | 300 | 51.26° N | 2022/5/8 19:18 | 771±134 | 867.5 | 925 | Day |
| South Africa | Medupi | 240* | 23.71° S | 2022/7/24 07:21 | 598±98 | 516.6 | 515.3 | Day |
| South Africa | Matimba | 240* | 23.60° S | 2022/7/24 07:21 | 708±118 | 617 | 664.6 | Day |

| | | | | | | | | |
|---|---|---|---|---|---|---|---|---|
| Russia | CHP-1 | 240* | 69.33° N | 2022/6/14 21:07 | 109.7±18 | 83 | -- | Night |
| Russia | CHP-3 | 240* | 69.32° N | 2022/6/14 21:07 | 57.1±12 | 44.1 | -- | Night |
| Russia | GRES-2 | 420 | 61.28° N | 2022/7/24 21:36 | 1287.3±143 | 1001.1 | -- | Night |
| Korean | Taean | 240* | 36.90° N | 2022/6/03 04:44 | 991.5±73 | 1022.2 | 900.5 | Day |
| Korean | Daesan | 240* | 36.99° N | 2022/6/03 04:44 | 30.4±3.4 | 23.9 | -- | Day |

* The default stack height is 240 m (Nassar et al., 2021)

21. Line 200: What is meant by: "OCO-2 satellites have no valid Gaussian plume data?

Answer: Due to the high latitude of this power plant, the small amount of passive remote sensing data and the low coverage of satellite data, we downloaded the OCO-2 satellite data for 2022-2024, which did not have enough 'good' overpasses to analyze the temporal differences in emissions from this power plant. We have revised in Line 223: "Due to the plant's high latitude, around 57° N, passive remote sensing satellite data are sparse (Insufficient OCO-2 overpasses were available for GPM at this plant), making it difficult to estimate."

22. Table 2: Please provide also the year of the respective overpass.

Answer: Revised

23. Figure 2: The source of the maps should be given. It would be nice to provide a map insertproviding a better information about where the GRES power plant is located. Please increase the axis labels on the left figure.

Answer: Revised.

[Figure]

24. Figure 3:   Please increase the figures and labels for better readability. It might be more instructive to show the overpasses in the same map cutout and scale. Is there a significance in the size of the wind arrows? I suggest to provide approximate wind speed information and local time of overpass. Probably is makes no sense to apply the same scale to all graphs due to different background values, but at least the same spread (2 ppm?) should be applied to all figures.

Answer:  Thanks for your suggestion. We have modified the figure 3 as you suggested. The size of the wind direction arrow is meaningless; the arrow point indicates the wind direction after interpolation of the ERA-5 data.

[Figure]

**Figure 3: Typical daytime and nighttime DQ-1 overpasses around the Reftinskaya GRES power station, with the red six-pointed star indicating the location of the power station and the arrow representing the wind direction at the plume.**

25. Line 216:   How is the lift height derived from stack height and atmospheric stability. Does the temperature of the gas play a role?

Answer:Thank you for your suggestion, as we have already explained in Section 2, the plume lift height is uncertain, we have assumed a lift height of 250m as a default value based on the study according to Nassar et al. (Nassar et al., 2021) and the uncertainty of this assumption is analysed later.

26. Line 253:   Provide a number for the linearly varying background.

Answer:We have revised in Line 274: "In this observation, the atmospheric background field exhibited a strong linear variation trend (About 0.015 ppm per kilometer along the track), and the Gaussian linear fitting results are shown in Fig. 6b, with an average $XCO_2$ background concentration of 428.9 ppm."

27. Figure 7:   Climate Trace (not Change?); The reader may think that this is continuous data, but the lines consist of 7? individual daytime and 9? nighttime observations. Please plot the data points!

Answer:Thanks for your comments, at the time the article was submitted, ClimateTRACE had not yet published the monthly resolution data, and so far, the official website has provided monthly emissions data up to November 2024 for each power plant. I have modified this graph, and labelled the data points as requested.

[Figure]

**Figure 7: Daytime (red) and nighttime (blue) CO₂ emission rates from GRES power plants (2022–2024). Solid lines represent modeled estimates with uncertainty bounds (shaded areas); red and blue dashed lines represent average daytime and nighttime emissions, and black dashed lines represent Climate TRACE emission inventory values.**

28. Line 298:   Which power plants provide stack heights? Include this information in one of the Tables.

Answer: We have added the corresponding stack heights for the power plants in Table 2 and labelled the plants for which no stack information was provided.

29. Line 300:   The authors must comment on the single pulse-pair precision that is used to fit the Gaussian plume. On short spatial scales, this information is more important than the accuracy of the IPDA measurement since the background is subtracted.

Answer: The results of the fitting in this study are the result of a 10-km moving average of the original single pulse echo signal, as explained above, and the results of the model simulations were similarly smoothed to ensure theoretical support during the least-squares fitting. After smoothing, the pseudo-observations are generated at the same 330-m interval, while the uncertainty of each of their footprints is less than 1 ppm, resulting in an average relative error of 47.3 % in the $XCO_2$ enhancement at the peak of the plume. We have revised in Line 362: "The uncertainty of $XCO_2$ around the studied power plants is less than 1 ppm by moving average, but the average relative error of $XCO_2$ enhancement at the peak of the plume is as high as 47.3%. Compared to the statistical uncertainties reported in Han et al (Han et al., 2024), both investigations identified that uncertainties in the DQ-1 satellite's $XCO_2$ observations dominate the error budget, accounting for approximately 50% of the total error."

30. Table 1/2:   Swap the order of the tables. Table 1 should contain the basic information about the power plants and Table 2 the uncertainties.

Answer: Thanks, we switched the order of sections 3.2 and 3.3 as you suggested, comparing the results with the emission inventories first, and analysing the uncertainties later.

31. Figure 8:   See comment above about the error of the data base.

Answer: We do not have access to the statistical errors for these specific emission inventories, so we are unable to include an error bar for the emission inventories in Figure 8. Because the emission inventories themselves are uncertain, we include cautionary language in the text to indicate that this is only a comparison with the emission inventories and is not representative of the true emissions. We have revised in Line 333: "It is critical to acknowledge that direct validation against stack monitor measurements is unavailable, and emission inventories are inherently uncertain and not independently validated. Therefore, comparisons between estimated results and inventories should be interpreted cautiously and serve as a provisional reference rather than definitive conclusions."

e

32. Line 325:   Without robust evidence on the diurnal (day-night) and seasonal (winter-summer) variations in power plant emissions, any claims about these differences are speculative and should be

clearly labeled as such.

Answer:Thank you for your suggestion, we have modified the expression in Line 323: "Overall, the $CO_2$ emissions predicted by the Gaussian plume model are generally higher than those in the emissions inventory. This discrepancy may be due to the timing of observations, as some were conducted during winter and summer in the Northern Hemisphere when increased electricity demand (e.g., air conditioning usage) prompts power plants to elevate generation capacity."

33. Line 333: Quantify the higher emission rates vs. Carbon Brief and Climate TRACE.

Answer:Revised in Line 331 :"Overall, Quantify the higher emission rates vs. Carbon Brief and Climate TRACE, a difference that might arise because conventional power plants undergo annual shutdowns for inspections, lowering annual averages relative to instantaneous emissions."

**Conclusions:**

34. In the conclusions, I would like to see suggestions for potential improvements in emission estimate accuracy, such as utilizing more advanced models or those with higher spatial resolution. For instance, the authors briefly mention WRF (Weather Research and Forecasting) model in the introduction. However, it would be helpful to provide a more detailed discussion on what enhancements can be expected from using this model instead of a Gaussian plume model, as well as any challenges that might arise.

Answer:The WRF-STILT model was used to estimate the assessment of urban emission sources, for point sources it is not suitable to use the WRF-STILT model, for Gaussian plume models, the wrf can provide refined small and medium scale meteorological fields, and I think the results will be more accurate than the results after interpolation by reanalysing the information such as the ERA5, especially in the case of a complex underlying surface with the influence of turbulence, and it would be better to use the WRF-LES model as the meteorological driving field for Gaussian model may be able to further improve the accuracy of the model and substantially reduce the model error due to wind field uncertainty. We have revised in Line 389: "Utilizing high-resolution wind fields simulated by the WRF-LES model around power plants to drive the Gaussian plume model may reduce uncertainties in wind field. Establishing automatic weather stations around the power plant for real-time monitoring of atmospheric radiation and surface wind speed could reduce errors caused by uncertainties in atmospheric stability."

**Data availability:**

35. Please give a hint about the availability of ACDL data, since this is the major data source used in this work.

Answer:Thank you for your advice. The ACDL data is not yet open for access, and researchers can download the satellite's data by making an official request through this website:

https://data.cresda.cn/#/home   We have also given the official link to the data and explained it in the article Data Availability.

"The ACDL dataset is under restricted access, the data can be requested at https://data.cresda.cn/#/home "

**Response to RC2:**

Authors: Xuanye Zhang, Hailong Yang, Lingbing Bu, Zengchang Fan, Wei Xiao, Binglong Chen, Lu Zhang, Sihan Liu, Zhongting Wang, Jiqiao Liu, Weibiao Chen and Xuhui Lee

The authors present a nice analysis of the nighttime and daytime power plant emissions at the GRES power plant in Russia using the novel DQ-1 lidar data. Overall the concept of the study is well-formed and will eventually be worthy of publication after some clarifications of the methodologies and revisions of the presentation as detailed below.

Answer: We greatly appreciate your valuable time for reviewing our research paper and providing suggestions. *(The **blue** text is in response to your comments, and the **green** text is for specific modifications in the paper)*

A major concern for this reviewer is the lack of access to the DQ-1 data on which the study is based.

Answer: Thank you for your advice. The ACDL data is not yet open for access, and researchers can download the satellite's data by making an official request through this website: https://data.cresda.cn/#/home We have also given the official link to the data and explained it in the article Data Availability. :

"The ACDL dataset is under restricted access, the data can be requested at https://data.cresda.cn/#/home "

**Specific Comments:**

Line 62: "OCO-2 is widely used..." for what purpose?

Answer: Thank you for your comment, I may have expressed myself inappropriately in the original article, what I wanted to express is that in the existing studies, it is common to use OCO-2 observations in conjunction with Gaussian plume modelling to estimate emissions from point sources. We have revised in Line 61: "The Orbiting Carbon Observatory-2 (OCO-2) has high measurement accuracy and stable results (Sheng et al. 2023, Crisp et al. 2017, Miller et al. 2007), and its $XCO_2$ product can be used in conjunction with GPM for the estimation of point source emissions."

Line 65: Even beyond quantifying emissions, Nassar et al (2017,2021) quantify uncertainties - these should be mentioned.

Answer: Thank you for your suggestion, we have already cited the uncertainty quantification of Naasar et al., and revised in Line 67: "Nassar et al. used the Gaussian plume model to estimate $CO_2$ emissions and uncertainties from 20 power plants

and related facilities in the U.S., India, South Africa, Poland, Russia, and South Korea, noting an average difference of 15.1 % between the estimated emissions and reported values for U.S. power plants (Nassar et al., 2021)."

Line 71: OCO-2 and OCO-3 have resolution 1.5km x 2.25km, GOSAT has resolution of 10km x 10km. They also sample differently, so this statement is a bit too glib of a comparison.

Answer:Thank you for your advice. The point I want to express here is that both OCO-2/3 and GOSAT are passive remote sensing detections, and the observation range receives the limitation of the solar altitude angle, thus the amount of data at high latitudes is very scarce. However, following your suggestion, since there is no case of using GOSAT observations in combination with Gaussian plume modelling, the text has been abridged. We have revised in Line 70: "OCO-2/OCO-3 are passive remote sensing satellites, which present data gaps in high-latitude and nighttime observations, and their spatial resolution of 1.5 km × 2.25 km poses limitations for monitoring small-scale strong point sources (Shi et al., 2023; Eldering et al., 2019)."

Line 78: I would replace "accuracy" with "uncertainty". Is the 1ppm number for a single 330m footprint? That isn't clear from how this is worded. Previous experience from the ASCENDS flight campaigns required at least a few kilometers of along track averaging to get the random errors below 1 ppm.

Answer:Thanks, my previous description was a bit unclear, the design index of DQ-1 is 1ppm accuracy at 50km resolution. 1ppm here means that the average deviation is less than 1 ppm after comparing the dq-1 measurement results with the TCCON site observation results. dq-1 has a footprint interval of 330 m, and by means of moving averaging, it can build up pseudo-observation sequences at 330m intervals, which is much smaller than the interval of the OCO-2 observation points, and then be applied to the Gaussian plume model, which can also capture small emission sources,, improving the capture rate of point sources. We have revised in Line 81: "The XCO$_2$ results from DQ-1 were validated against TCCON observations, showing an average deviation of less than 1 ppm at a 50 km (149 footprints averaged) resolution (Zhang et al., 2024). Additionally, its satellite footprint interval is 330 meters, significantly enhancing the estimation accuracy for small power plants. (Zhang et al., 2023)."

Section 2.1.1: I believe this section needs a bit more detail on the retrieval of XCO2 from the lidar. This is not a technology that many readers will be familiar with and so deserves a bit more detail regarding differential optical depths and weighting function correction. Are you correcting for water vapor (which is also released by power plants), etc? Even though the retrievals are not the focus of this paper, the data quality is a big concern and that is not addressed here.

Answer: We have included an inversion method for XCO2 in Section 2.1.1, which will enable readers who have not been acquainted with IPDA lidar to understand the data processing principles involved, and for water vapour, we have taken into account the mixing ratio of the water vapour when calculating the number density of atmospheric molecules in the integral weighting function, and the differential absorption cross section of the water vapour has little or no effect due to the close proximity of the ON-LINE and OFF-LINE wavelengths . We make the following additions in Section 2.2.1 of the article: "XCO$_2$ can be calculated using these two wavelength echo signal strengths combined with following equation (Ehret et al., 2008):

$$XCO_2 = \frac{DAOD}{IWF} = \frac{\frac{1}{2}ln\frac{P_{off}E_{on}}{P_{on}E_{off}}}{\int_{P_{TOA}}^{P_G}\frac{\Delta\sigma(p,T)}{(m_{dryair}+\rho_{H_2O}(p)m_{H_2O})g}dp} \tag{1}$$

*DAOD* is Differential Absorption Optical Depth, *IWF* is the integral weight function, $P_{on}/P_{off}$, represents the echo power of the two laser beams, $E_{on}/E_{off}$ represents the emitted power, $p_{TOA}$ and $p_G$ are the pressure at the top of the atmosphere and at the ground, $\Delta\sigma(p,T)$ represents differential absorption cross-section, $m_{dryair}$ is the molecular mass of dry air, $\rho_{H_2O}$ is the relative humidity of the air."

Section 2.1.2: It's not clear how the wind data is being created here. Are you using the full 3D winds and interpolating them to 240 + 250m? You mention the use of "ground-level" winds, but I'm not sure if you mean the surface or the planetary boundary layer. Perhaps a figure would be helpful to show how you use the model fields to create the steady state wind value for the GP model. Models are also notoriously bad at nighttime PBL depths. Did you evaluate these wind fields against any atmospheric data, especially at night?

Answer: Thanks for your suggestions! The high-altitude wind data mentioned here, with reference to Nassar et al. (Nassar et al., 2021), directly interpolates the three-dimensional wind field of ERA-5 to the height of 240+250 m. The ground winds here are surface winds(2-m), which I may not have expressed very clearly, and have been modified to the surface 2 m wind speeds, which are mainly for the subsequent calculations of the parameters of atmospheric instability. We have revised in Line 123: "In this study, atmospheric instability is calculated from surface wind speeds and cloud cover, with surface wind speed data selected from spatially interpolated ERA5-land hourly wind speed U and V vectors."

The specific three-bit wind field interpolation method can be divided into two steps, the specific method is shown in the following figure:

[Figure]

We do not have specific real measurements, but only comparisons between data from two wind fields, which are themselves subject to uncertainty, so we take this into account when calculating the model error. We also give some points in our conclusions that show that the error due to the uncertainty of the wind field can be significantly reduced if ground-based

meteorological observatories are established around the power plant. We revised in Line 387: "he results show that during the daytime, the error in the surface wind field is higher due to turbulence, which can cause some invalid observations or increase the error caused by atmospheric instability and wind field to the model. Utilizing high-resolution wind fields simulated by the WRF-LES model around power plants to drive the Gaussian plume model may reduce uncertainties in wind field. Establishing automatic weather stations around the power plant for real-time monitoring of atmospheric radiation and surface wind speed could reduce errors caused by uncertainties in atmospheric stability."

Equation 3: is there a mismatch between this and Equation 1? How are the mean and standard deviation and the parameter a specified?

Answer: The parameter a in Equation 1 represents the atmospheric instability parameter, and the parameter a in Equation 3 represents the fitting parameter in the process of Gaussian linear fitting of the raw data by least squares, and in order not to mislead the readers, I have replaced the a in Eq. (3) with A,and explained it. We also revised in Line 155: "

$$XCO_2(x) = XCO2_b + B \cdot x + \frac{A}{\sigma\sqrt{2\pi}} e^{[-(x-\mu)^2/2\sigma^2]} \tag{4}$$

Where $A, B, \sigma, \mu$ are the parameters in the linear fit function, obtained by least squares fit, $XCO2_b + B \cdot x$ is background value of $XCO_2$, $\frac{A}{\sigma\sqrt{2\pi}} e^{[-(x-\mu)^2/2\sigma^2]}$ is $\Delta XCO_2$ caused by power plant emissions (Reuter et al., 2019)."

Line 158-161: Can you provide a reference for the interpolation? Is there an equation you're working with? For this process, are you simultaneously optimizing the CO2 emissions and the stability parameter at the same time?

Answer: Thank you for your comments! In this paper, the atmospheric instability is calculated using the method of Gordon et al. after the Pasquill classification method is specified (e.g., Table S1/S2/S3), and the net radiation index is calculated by using the total cloud amount, low cloud amount, and other data in the ERA-5 data combined with the solar altitude angle, and then combined with S2/S3, and then the preliminary atmospheric instability is obtained by linear interpolation based on the surface wind speed. The parameter a1 is then calculated from the standard deviation of the surface wind speed, and then the parameter a and the CO2 emission are optimised simultaneously to find the optimal atmospheric instability parameter a2 within the interval [a1-sigma,a1+sigma], and then the corresponding atmospheric instability parameter brings about the uncertainty $\varepsilon_a = Q(a2) - Q(a1)$

$$\sigma_y(x) = a \cdot (x/1000)^{0.894} \tag{1}$$

Table S1

| Total cloud cover/low cloud cover(0-1) | Net Radiation Index | | | | |
|---|---|---|---|---|---|
| | nightime | solar altitude angle | | | |
| | | ≤15° | 15°-35° | 35°-65° | > 65° |

| | -2 | -1 | +1 | +2 | +3 |
|---|---|---|---|---|---|
| ≤0.4/≤0.4 | -2 | -1 | +1 | +2 | +3 |
| 0.5-0.7/≤0.4 | -1 | 0 | +1 | +2 | +3 |
| ≥0.8/≤0.4 | -1 | 0 | 0 | +1 | +1 |
| ≥0.5/0.5-0.7 | 0 | 0 | 0 | 0 | +1 |
| ≥0.8/≥0.8 | 0 | 0 | 0 | 0 | 0 |

Table S2

| Surface 2-m wind speed (m/s) | Net Radiation Index (Table S1) | | | | | |
|---|---|---|---|---|---|---|
| | 3 | 2 | 1 | 0 | -1 | -2 |
| <2 | A | A-B | B | D | (E) | (F) |
| 2-3 | A-B | B | C | D | E | F |
| 3-5 | B | B-C | C | D | D | E |
| 5-6 | C | C-D | D | D | D | D |
| >6 | C | D | D | D | D | D |

Table S3

| stability parameter | A | B | C | D | E | F |
|---|---|---|---|---|---|---|
| Parameter a | 213 | 184.5 | 150 | 130 | 104 | 77.6 |

We have also revised in Line 174: "In this study, empirical interpolation of atmospheric stability parameters was implemented by accounting for surface wind speed, cloud coverage, and solar elevation angle, the latter calculated from latitude and time of observation (Nassar et al., 2021). The uncertainty in the stability parameter was quantified through the uncertainty in surface wind speed measurements. Subsequently, a least-squares fitting approach was applied under the assumption that prior estimates of atmospheric stability parameters varied within one standard deviation. The optimal solution was then selected as the representative atmospheric instability value for the target location."

Line 161-162: What sort of smoothing? What are the length scales?

Answer:Thank you for your comments! We used a moving average smoothing method, where each grid point averaged 10 km around it, corresponding to 30 pluses, by which the uncertainty of XCO2 was made less than 1 ppm, and thus the XCO2 enhancement signal could be detected. As shown in the following figures, the original monopulse echo signal of a set of

overpasses, and the pseudo-observation sequence after 10-km sliding average are shown, and the histograms of the precision distributions corresponding to the single-pulse signal and the XCO2 after moving average are computed.

[Figure]

Figure 1. The figure on the left shows the DQ-1 single-pulse echo signal. The right figure shows the XCO2 after 10-km moving average, and the trend of XCO2 can be clearly seen in the right figure.

[Figure]

Figure 2. The left figure shows the accuracy corresponding to the DQ-1 single-pulse echo signal. The right figure shows the accuracy of XCO2 after 10-km moving average.

We also have explained in Line 179: "A 10-km moving averaging was applied to DQ-1 data, reducing the uncertainty of $XCO_2$ to below 1 ppm, which facilitated the detection of enhanced $XCO_2$ signals (Zhang et al., 2023)."

Line 172-173: This is a bit unclear - are you using the inferred emission rates from the two models' wind fields to compute the errors? Similarly for the other error terms? What is the 1-sigma parameter uncertainty in your stability parameterization and how is it calculated?

Answer:The uncertainty of all parameters in this paper is calculated by taking all possible values of the parameter into the model and calculating a set of $CO_2$ emissions, and then calculating the standard deviation of this set of results, with 1-sigma as the uncertainty that the parameter introduces into the model. For example, for wind speed, since the wind speed of ERA-

5 is used as the true value in this paper, the co2 emission $Q_{ERA-5}$ is calculated, and the wind speed of MERRA-2 is also brought into the model to calculate the corresponding emission $Q_{MERRA-2}$. The uncertainty caused by the wind speed is:

$$\varepsilon_s = \sqrt{\frac{(Q_{ERA-5} - Q_{MERRA-2})^2}{2}}$$

We have also revised in Line 191: "Where $\varepsilon_s$ represents the error caused by wind speed. This is estimated by comparing the $CO_2$ emissions from the target power plant using wind speeds interpolated from MERRA-2 and ERA-5 data, with the wind speed uncertainty given by the standard deviation between the two predictions."

Section 3.1: This analysis is very interesting, but a bit hard to follow. I suggest you separate this section into subsections by dates of overpass and provide a table with the different inferred emission rates for all of the 19 days and the variations in posterior errors. Perhaps this is the information contained in Figure 7? Can you adjust the figure caption to specify the meaning of the shading?

Figure 2: please cite the source of these images.

Answer: The shaded areas represent the range of error in the model estimates, and I have table all of the results for the 19 days in Table 1 of the text, which I also explain in more detail in the caption to Figure 7: "Figure 7: Daytime (red) and nighttime (blue) $CO_2$ emission rates from GRES power plants (2022–2024). Solid lines represent modeled estimates with uncertainty bounds (shaded areas); red and blue dashed lines represent average daytime and nighttime emissions, and black dashed lines represent Climate TRACE emission inventory values. "

Table 1 Emission estimates and uncertainty for GRES power plant

| Date | Day or Night | Estimated emissions (kg s⁻¹) | Climate TRACE (kg s⁻¹) | Estimated Uncertainty (kg s⁻¹) |
|---|---|---|---|---|
| 2022-08-17 | Night | 966 | 697.1 | 156.5 |
| 2022-10-23 | Day | 1018.5 | 748.1 | 186.9 |
| 2022-11-20 | Night | 1029.9 | 772 | 150.8 |
| 2023-02-09 | Day | 1157.6 | 815.2 | 153.1 |
| 2023-03-30 | Night | 958.5 | 892.4 | 130.9 |
| 2023-04-01 | Day | 918.6 | 772.8 | 142.6 |
| 2023-05-20 | Night | 897.5 | 746.1 | 100.4 |
| 2023-05-22 | Day | 876.5 | 746.1 | 173.5 |
| 2023-06-23 | Night | 764.4 | 725.3 | 118.9 |
| 2023-07-24 | Night | 887.6 | 748.8 | 83.0 |

| | | | | |
|---|---|---|---|---|
| 2023-09-08 | Day | 951.5 | 733 | 147.4 |
| 2023-10-17 | Night | 957.8 | 808.6 | 139.8 |
| 2023-11-05 | Day | 988.6 | 834.1 | 131.5 |
| 2023-12-10 | Night | 1177.6 | 904.3 | 169.2 |
| 2024-02-01 | Day | 1158.6 | 844.1 | 187.6 |
| 2024-02-08 | Day | 1105 | 844.1 | 150.4 |
| 2024-03-17 | Night | 1070.5 | 892.4 | 168.7 |
| 2024-06-24 | Night | 819.7 | 725.3 | 93.6 |
| 2024-07-01 | Night | 874.5 | 748.8 | 126.8 |

Line 222: "The slightly higher result..." - the lower result in the inventory is well within 1-sigma of the DQ-1 informed estimate. It could just be due to random error from wind speeds, etc

Answer:Thanks for the correction. This result really shouldn't be emphasised as being higher than the emission inventory, it's within the estimation error, and the point I'm trying to express here is that the fact that the emission inventory expresses the total annual emissions, and then we convert it to a per-second emission result, can lead to a low result for the emission inventory compared to the instantaneous emissions. We have modified this sentence in the text in this section and in Line 291 of the text to reflect the opinion I want to express: "The plant undergoes annual shutdowns for maintenance, and the satellite observations represent instantaneous emissions, which may differ slightly from the annual average emissions."

Line 225: "...lower operational efficiency...resulting in CO2 emissions exceeding the inventory" - wouldn't this situation produce less CO2 and more CO emissions?

Answer:Thanks for the comments. Although incomplete combustion may increase CO production, the total $CO_2$ emissions per unit of electricity generated will still increase due to the need to burn more coal and oil as a result of incomplete combustion. And, at low loads, the operation of power generating equipment with frequent starts and stops can also lead to a reduction in the efficiency of power generation, with the overall result that more coal or oil needs to be burned to generate the same amount of electricity, thus producing more $CO_2$ (Hendriks, 2012). I have not been very clear here, which has led to a misunderstanding on the part of the reader, and have made a corresponding change in Line 248: "In mild summers, despite reduced nighttime electricity demand and plant output, low-load operations impair combustion efficiency, increasing fuel use per kWh and exacerbating $CO_2$ emissions through frequent start-stop cycles (Hendriks, 2012)."

Line 237: It's a small nit, but earlier you said you excluded tracks that were > 30km from the source.

Answer:The actual data filtering we performed was 50 km, and this filtering scheme was also reflected in line 160 of the original article: "For a power plant with an emission rate of 1000 kg s$^{-1}$, the downwind $XCO_2$ enhancement at 50 km under 10 m s$^{-1}$ winds is <1 ppm, which is less than the uncertainty of $XCO_2$ observed by the DQ-1, indicating low reliability in distant plume detection. Therefore, to improve the model's fit with satellite observation results, the selected DQ-1 orbital data require that the downwind direction of the point source intersects with the satellite footprint and that the distance between the $XCO_2$ enhancement location and the point source is less than 50 km." line 148 line > 30km for the expression of the error, 30-km has been revised to 50km, for 1000Kg / s emission rate of the power plant, at a wind speed of 10m / s, the delay plume downwind of 50km, $\Delta XCO_2$ is less than 1 ppm, less than the uncertainty of $XCO_2$, we believe that the results at a distance away from this distance must be inaccurate, therefore, this condition was added to the raw data filtering.

Line 279: Unauthorized emissions are mentioned a few times in the paper, but it's not clear what the regulations are and who is enforcing them. Is there a regulation in Russia about CO2 emissions at night vs. in the daytime?

Answer:According to the bill proposed by the Russian government in 2019 (LOC, 2019), which focuses on the development of national emission inventories and sectoral emission reduction targets, there is no explicit requirement for differential monitoring by time of day (e.g., night vs. day). In actual implementation, emissions accounting is based on annual or quarterly averages rather than hourly-level data (Liu et al., 2023). Unauthored emissions in the article refers to emissions that far exceed the results of the emission inventory statistics, however, the estimated CO2 emissions based on the statistical observations in this study do not far exceed the emission inventory, i.e., there is not much difference between the reported data and the actual monitoring data of this power plant.

Line 293-298: Is the uncertainty on atmospheric stability well-defined, since you are fitting for it in your retrieval?

Answer:We have made some changes in line 22 based on the comments you made: "$\varepsilon_a$ represents the error induced by atmospheric instability, which is quantified using surface wind speed and net radiation index. The uncertainty in atmospheric instability is derived from the uncertainty in surface wind speed data."

Section 3.2: This is a key section, but there are no comparisons to previous findings. How do your uncertainties compare to work by Nassar and others?

Answer:Our study incorporates slightly different error components compared to prior work by Nassar and Guo et al., who quantified uncertainties from secondary emission sources, wind fields, plume rise height, and background concentrations with their analyses showing wind field uncertainty dominating total emission uncertainty (>60%). However, DQ-1's active lidar detection method, characterized by sparse plume sampling (lacking spatial swath coverage), introduces significant observational uncertainties that disproportionately affect background $XCO_2$ determination. Consequently, we explicitly

integrated XCO₂ observational uncertainty into our model uncertainty analysis. Results reveal that background concentration uncertainty contributes 40.7%, wind field uncertainty 26.7%, and atmospheric instability uncertainty (driven by surface wind speed errors) 25.1%. These findings align with Han et al.'s satellite-based study (49.7% observational uncertainty contribution). We have also revised in Line 364: "Compared to the statistical uncertainties reported in Han et al (Han et al., 2024), both investigations identified that uncertainties in the DQ-1 satellite's XCO₂ observations dominate the error budget, accounting for approximately 50% of the total error. Beyond this, the significant contribution of wind field uncertainties aligns with findings from Nassar et al (Nassar et al., 2017; Guo et al., 2023). In contrast to previous studies, this work incorporates uncertainties in atmospheric instability. Due to the influence of turbulence and other factors within the boundary layer, the uncertainty in surface wind speed also exerts a significant influence on atmospheric instability calculations."

Section 3.3: This is not really a "validation" of emissions, but rather just a comparison with the inventories. Some of your conclusions here are speculative and need to be augmented with the appropriate caveats. Right now they read as unsupported statements without citations. Are there any stack monitors at these power plants?

Answer:These power plants have not been able to find stack detectors, and we are really only comparing with emission inventories, and have changed the title to 'Comparison with emission inventories'. Because emission inventories are inherently uncertain and no conclusions can be drawn from a comparison with emission inventories, I have added a cautionary note to the Line 333: "It is critical to acknowledge that direct validation against stack monitor measurements is unavailable, and emission inventories are inherently uncertain and not independently validated. Therefore, comparisons between estimated results and inventories should be interpreted cautiously and serve as a provisional reference rather than definitive conclusions."

Figure 8: I think your axis labels have the wrong units - do you mean kg/h?

Answer:Thanks, revised.

**Summary:**

To summarize, the manuscript presents valuable findings based on new data, but also exhibits some shortcomings, including missing citations and a critical oversight regarding the impact of turbulence on the  results. Notably, this omission is particularly concerning in light of potential differences between daytime and nighttime measurements. In order to ensure the manuscript's credibility and thoroughness, I believe it is  essential to address these issues before publication.

Answer: We greatly appreciate your valuable time for reviewing our research paper and providing suggestions. We have revised the manuscript according to your comments point-to-point